# Attributing the decadal variations in springtime East Asian and North American dust emission to regime shifts in extratropical cyclone

**Yiting Wang[1], Yan Yu[1,2,3], Ji Nie[1,2,3], and Bing Pu[4]**

[1]Department of Atmospheric and Oceanic Sciences, School of Physics, Peking University, Beijing, China
[2]Laboratory for Climate and Ocean-Atmosphere Studies, Department of Atmospheric and Oceanic Sciences, School of Physics, Peking University, Beijing, China
[3]China Meteorological Administration Tornado Key Laboratory, Beijing, China
[4]Department of Geography and Atmospheric Science, University of Kansas, Lawrence, KS, USA

**Correspondence:** Yan Yu (yuyan@pku.edu.cn) and Ji Nie (jinie@pku.edu.cn)

**Abstract.** Dust activities across East Asia and North America have shown decadal variations, mediating radiation budget, air quality, and human health, especially during their peak months of April and May. Using satellite and ground measurements, as along with simulations from a dust emission model, we demonstrate an increase of 12.7 % and 23.4 % in April dust emissions across East Asia and North America, respectively, during the past four decades, in contrast to a 16.5 % and 2.5 % decrease during the last two decades. Meanwhile, both regions show a steady increase in May dust emissions by 5.7 % and 16.3 %, respectively, since the 1980s. Sensitivity experiments attribute both regions' decadal variations in dust emission primarily to surface wind speed changes; whereas vegetation exerts minimum influence on the regional dust emission variations. Furthermore, these decadal variations in dust initiating wind could largely be attributed to regime shifts in extratropical cyclone (EC), including their duration and intensity. Specifically, ECs are responsible for 60 %–70 % of the April–May total dust emissions in East Asia and 30 %–40 % of that in North America; meanwhile, ECs explain a larger portion of the decadal variations in April dust emission from East Asia (up to $\sim 80$ %), compared with May and from North America. These results highlight the changing frequency and duration of strong winds, especially those associated with EC, and their role in shaping the decadal variations of mid-latitude dust emissions.

## 1 Introduction

East Asia and North America are significant dust source regions in the Northern Hemispheric mid-latitudes. The presence and transportation of dust alter the radiation budget (Adebiyi et al., 2023; Chen et al., 2013; Huang et al., 2014) and biogeochemical cycles of the marine and terrestrial ecosystems (Jickells and Moore, 2015; Kong et al., 2022; Jickells et al., 2005). Apart from the influence on the natural environment, extreme dust activities from the Gobi Desert and Southwest United States also impair atmospheric visibility, air quality, and human health across downwind regions, including populated areas in China and the United States

(Gui et al., 2022; Hashizume et al., 2020; Yang et al., 2015). These environmental and societal concerns of dust activity peak in April and May across both regions, when vegetation and snow provide insufficient protection of the dry soil, accompanied by strong near-surface winds and frequent extratropical cyclones (Aryal and Evans, 2022; Kim et al., 2017, 2021; Kurosaki and Mikami, 2007). Changes in these atmospheric and land surface factors also shape the interannual and decadal variations in springtime dustiness across these two mid-latitude regions, ultimately affecting the regional human well-being.

Springtime dustiness across both East Asia and North America exhibits substantial interannual-to-decadal varia-

tions, with seemingly opposing trends across decades and divergent driving mechanisms as reported by various studies (Kim et al., 2021; Kurosaki and Mikami, 2007; Xu et al., 2006). Based on satellite measurements of vegetation greenness and dust aerosol abundance, Wang et al. (2021) explained the decreased East Asian dustiness in spring by ecological restoration and resultant vegetation expansion for the period 2001–2020. In contrast, Wu et al. (2022) performed simulations with a dust emission model and identified surface wind speed weakening as the dominant driving mechanism for the decreased springtime dust emission across East Asia since 2000s. Apart from that, Song et al. (2021) regarded increasing vegetation and decreasing surface wind as main contributors of decreasing East Asian dust optical depth during 2009–2019. While this identified role of wind speed confirmed Tai et al.'s (2021) findings based on chemical transport model simulations, the latter study, covering the longer period since the 1980s, showed a different phase of decadal variations in East Asian dust emissions (Tai et al., 2021). Similar debates regarding the interannual-to-decadal variations in North American dustiness have also persisted. For example, statistical analysis with ground-based surface fine dust revealed warming and drying as the main cause of North American springtime dust emission's increase in the early 21st century (Achakulwisut et al., 2017). While according to ground-based dataset in the long-term, Pu and Ginoux (2018) attributed increased springtime surface fine dust concentrations to the decreasing trend in precipitation across Southwestern America for the period 1990–2015. Similar spatiotemporal variation was interpreted by Hand et al. (2017) as a result of wind speed, soil moisture and land cover changes. In contrast, Pu and Ginoux (2017) studied satellite derived dust optical depth and established the primary relevance between land cover change and Southwestern North American springtime dust activities for the period 2004–2015. In the context of global warming, both regions' dust emissions appeared sensitive to vegetation expansion, according to global climate models and coupled dynamic vegetation-chemical transport model simulations (Li et al., 2021; Pu and Ginoux, 2017; Zong et al., 2021).

Despite the rich body of literature on the interannual to decadal variations in springtime mid-latitude dustiness, little consensus has emerged among these model- and observation-based studies regarding the direction of change and underlying mechanism. Observational datasets from ground- and satellite-based measurements provide estimates for atmospheric dust load in the past two to four decades but lack dust emission records. Therefore, the observed increase or decrease in atmospheric dust load could be sourced from both local and remote emissions, especially for the highly transportable mid-latitude dust (Yu et al., 2019b). On the other hand, dust emissions could be quantified using models, but the credibility of these models, especially their sensitivity to different influencing factors, should be validated against observation. Moreover, intense dust storm events that frequently occur in April–May over the Gobi Desert and Southwest United States are often modulated by extratropical cyclones, and associated storm tracks or frontal systems (Lukens et al., 2018; Guo et al., 2017). For instance, the extreme dust event over East Asia in March 2021 was attributed to both the increased intensity and frequency of Mongolian cyclones (Liang et al., 2022; Yin et al., 2022). Between 2001–2022, Mongolian cyclones were reported to contribute 34 % to 47 % of the total dust emissions from the Gobi Desert (Mu and Fiedler, 2025). However, quantitative analysis of how the characteristics of extratropical cyclones, including their occurrence, intensity and size, affect long-term variations in springtime dust activity across East Asia and North America remains lacking, which limits our understanding of the drivers behind dust's interannual to decadal variability.

This study aims to reconcile the interannual to decadal variations in dust emission across East Asia and North America and quantify the influence of multiple environmental factors. In this work, we simulate dust emissions across East Asia and North America in April and May from the late 20th century to the early 21st century using an observation-validated dust emission model and subsequently quantify the contribution of meteorology and land surface factors on dust emission changes, including surface wind speed, soil moisture, snow cover fraction, surface temperature, leaf area index (LAI) from reanalysis and satellite-based observation datasets. Furthermore, this study integrates cyclone tracking and identification techniques to quantify the impact of extratropical cyclones (ECs) on the interannual to decadal variations of both regions' dust activity over the past four decades. To clarify the reliability of the off-line dust emission model based on Ginoux et al. (2012), we compare the simulated changes in dust emission with ground-based dust measurements, including the global dust Integrated Surface Database (duISD) during 1980–2019, and the Interagency Monitoring of Protected Visual Environments (IMPROVE) network during 1988–2021.

## 2  Data and method

### 2.1  Ground-based dust measurements

The observed extinction coefficient contributed by dust aerosol ($\beta$, km$^{-1}$) across East Asia (35–50° N, 90–120° E) is provided by global dust Integrated Surface Database (duISD) covering the period 1980–2019 (Xi, 2021). This dataset compiles about 30 000 stations globally, as collected by the National Oceanic and Atmospheric Administration (NOAA), and derives dust extinction coefficient from visibility observations as follows:

$$\beta = \frac{3.9}{V} \times f, \tag{1}$$

where $\beta$ is a measure of the extinction coefficient caused by dust particles, $V$ is the harmonic mean visibility associated with dust events, and $f$ is the dust frequency (%) given by:

$$f = \frac{N_{du}}{N_{ww}} \times 100\%, \qquad (2)$$

Here, $N_{du}$ is the number of reported dust events, and $N_{ww}$ is the total number of weather reports (ww) during a given time period (Shao et al., 2013; Kurosaki and Mikami, 2003). Weather reports from manned stations are categorized by the World Meteorological Organization (WMO) under Code Table 4677, with priority codes ranging from 00 (lowest) to 99 (highest), indicating the visual perception of weather phenomena during the observation period. Dust events are ranked within the fog (40–49) and precipitation (50–99) weather groups and are identified by the following numeric codes: ww = 06–09, 30–35, 98.

The calculation of $\beta$ is based on the principle that visibility is typically determined by light attenuation measurements using sensors such as transmissometers or forward-scatter sensors. We analyze long-term observations from 100 and 65 stations in East Asia for April and May, respectively, with valid records spanning over two years for both the late 20th century (1980–1999) and early 21st century (2000–2019).

Parallel to the duISD dataset, the Interagency Monitoring of Protected Visual Environments (IMPROVE) network has monitored the surface fine dust concentrations ($\mu g\,m^{-3}$) across North America (30–50° N, 103–118° W) since 1988. The IMPROVE network was originally designated to support the United States Environmental Protection Agency's Regional Haze Rule (Hand et al., 2019), and has subsequently been applied to air quality studies, including those on fine dust variations near the surface (Kim et al., 2021; Pu et al., 2022; Tong et al., 2017). The IMPROVE dataset provides individual species' contributions to $PM_{2.5}$ mass and total aerosol extinction twice a week during 1988–2000 and every third day after 2000 in the United States (Pu et al., 2022). In this work we analyze the surface fine dust concentrations ($\mu g\,m^{-3}$) from 1988 to 2021 in April and May.

## 2.2 Satellite-based dust measurements

To geographically constrain the off-line dust emission calculation (Sect. 2.6) to observed dust emission hotspots, here we analyze the Moderate-resolution Imaging Spectroradiometer (MODIS) with collection 6.1, level 1 provides the daily dust optical depth (DOD) during 2000–2021 at a spatial resolution of $0.1° \times 0.1°$. MODIS DOD is calculated from aerosol optical depth (AOD) and the Ångström exponent ($\alpha$) as follows:

$$DOD = AOD \times \left(0.98 - 0.5089\alpha + 0.051\alpha^2\right). \qquad (3)$$

The MODIS instrument is carried by both the Terra (equatorial overpassing at 10:30 LT) and Aqua (equatorial overpassing at 01:30 LT) satellites. DOD from MODIS is broadly used in the study of dust emission and atmospheric loading (Ginoux et al., 2012; Wu et al., 2022; Yu et al., 2019a; Yu and Ginoux, 2021, 2022; Meng et al., 2025) and widely provides the observational basis for dust emission simulation (Ginoux et al., 2012; Parajuli et al., 2019; Pu et al., 2020).

## 2.3 Satellite-based vegetation measurements

The long-term global leaf area index (LAI) is provided by Global Inventory Modeling and Mapping Studies LAI product (GIMMS LAI4g) (Cao et al., 2023), with a half-month temporal resolution and a spatial resolution of $1/12°$ for the period 1982–2020. In this study, we expand the time range to 1980–2021 by replacing LAI in 1980–1981 with that in 1982 and LAI in 2021 with that in 2020. The GIMMS LAI4g product used the PKU GIMMS normalized difference vegetation index product (PKU GIMMS NDVI) and high-quality global Landsat LAI samples to remove the effects of satellite orbital drift and sensor degradation of Advanced Very High Resolution Radiometer (AVHRR). The algorithm of compiling LAI also utilizes vegetation type reference from the MODIS Land Cover Type Product (MCD12Q1, version 6.1).

## 2.4 Reanalysis data

To investigate the change in dust emissions and the contribution of several environmental variables in April and May, we analyze the 6-hourly snow cover fraction (%), top layer soil moisture ($0$–$7\,cm\,m^3\,m^{-3}$ TS1), land surface temperature (K) and hourly 10 m wind speed ($m\,s^{-1}$) from the European Centre for Medium-Range Weather Forecasts Reanalysis v5-Land (ERA5-LAND, referred to ERA5 hereafter) during 1980–2021. The ERA5-LAND dataset is an enhanced global dataset produced by the European Centre for Medium-Range Weather Forecasts (ECMWF), with a native resolution of 9 km (Hersbach et al., 2020). The 10 m wind speed from ERA5 can capture the characteristics of wind to explore wind events both in the hourly and daily scales, compared with station observed wind speed from Hadley Centre's Integrated Surface Database (HadISD) (Molina et al., 2021).

## 2.5 Extratropical cyclone detection and tracking

To analyze regime shifts of extratropical cyclones and their contribution to near-surface strong winds ($> 6\,m\,s^{-1}$), we employ the Cyclone TRACKing framework (CyTRACK), an open-source Python toolbox for cyclone detection and tracking in reanalysis datasets (Pérez-Alarcón et al., 2024). CyTRACK identifies cyclone centers from mean sea level pressure (MSLP) fields at each time step and applies threshold-based filtering to track each cyclone. Previous evaluations have demonstrated that CyTRACK reliably reproduces interannual and seasonal variability, life-cycle characteristics, and spatial distributions of cyclone tracks when compared

with ERA5-based best-track archives and other cyclone-track datasets.

In this work, we use the 6-hourly 10-m wind speed ($\mathrm{m\,s^{-1}}$) and MSLP data in April and May from ERA5 to identify and track ECs during 1980–2021, with a horizontal resolution of 0.25°. Cyclone centers are mainly identified based on two criteria: (1) surface relative vorticity exceeding $10^{-5}\,\mathrm{s^{-1}}$, a threshold widely applied in EC detection studies (Chen and Di Luca, 2025; Chen et al., 2022; Priestley et al., 2020), and (2) the central MSLP anomaly being at least 1 hPa lower than the surrounding grid points (Eichler and Higgins, 2006). Only cyclones with a lifetime longer than 24 h are retained.

To quantify the contribution of ECs to surface wind speed across East Asia and North America, we define all surface wind speeds and strong-wind ($> 6\,\mathrm{m\,s^{-1}}$) events that occur within the radial domain of each extratropical cyclone as cyclone-affected winds and cyclone-affected strong winds. Conversely, winds and strong winds outside this domain are classified as non-cyclone-affected winds and non-cyclone-affected strong winds, respectively. The spatial extent of each cyclone is determined following Schenkel et al. (2017) as the radial distance from the cyclone center at which the azimuthal-mean 10 m wind speed equals a critical wind speed threshold. Following previous studies (Pérez-Alarcón et al., 2021, 2024), we test several thresholds (2, 4, 6, 8, 10, and 12 $\mathrm{m\,s^{-1}}$) and adopt $6\,\mathrm{m\,s^{-1}}$, which both aligns with our definition of strong winds and provides the most consistent results. All points within this radius are considered to be influenced by the cyclone.

## 2.6 Off-line dust emission model

The quantification of historical, springtime dust emission change across East Asia and North America is achieved by an off-line dust emission model, based on Ginoux et al. (2001, 2012). Dust emission flux $F_p$ is calculated as follows:

$$F_p = C S u_{max}^2 (u_{max} - u_t). \tag{4}$$

In April and even occasionally in May, mid-to-high latitude dust source regions in East Asia and North America are partly covered by snow or frozen soil, which has a nonnegligible influence on dust emission (Yin et al., 2022; Balkanski et al., 2004). In this work, we take snow cover faction, surface temperature and vegetation cover into consideration, and define the simulated dust emission flux $F_{p\text{-cover}}$ with $0.1° \times 0.1°$ spatial resolution as follows:

$$F_{p\text{-cover}} = F_p \times (1 - f_{snow}) \times \exp(-1 \times \mathrm{LAI}) \times I, \tag{5}$$

where $f_{snow}$ is daily snow cover fraction, LAI is daily vegetation cover and $I$ is the indicator function of surface temperature (surface temperature $> 0\,°\mathrm{C}$, $I = 1$; surface temperature $< 0\,°\mathrm{C}$, $I = 0$). $C = 1.9\,\mathrm{\mu g\,s^2\,m^{-5}}$ is a dimensional factor, $S$ is the fraction of dust source (Ginoux et al., 2010, 2012), approximated by the frequency of DOD $> 0.2$ for the period during 2000–2021 from MODIS in April and May, separately.

$u_{max}$ is daily maximum surface wind speed in the original model and is substituted with hourly 10 m wind speed in the current study. $u_t$ is the threshold wind velocity which is calculated as follows:

$$u_t = A \times u_{ref} \sqrt{\frac{\rho_p - \rho_a}{\rho_a} g \Phi_p} \left(1.2 + 0.2 \log_{10} w\right) (w < 0.5), \tag{6}$$

where $A = 6$ is a dimensionless parameter, $u_{ref}$ is a reference threshold wind speed from Pu et al. (2020). $\rho_a$ and $\rho_p$ are the air and particle density, $g$ is the gravitational acceleration, $\Phi_p$ is the particle diameter in five bins: 0.1–1, 1–1.8, 1.8–3.0, 3.0–6.0 and 6.0–20.0 μm, according to Kok et al. (2017), $w$ is the top-layer soil moisture ($\mathrm{m^3\,m^{-3}}$).

To assess the reliability of the off-line dust emission model over East Asia and North America during April and May, spatial distributions and temporal correlations between simulated dust emissions and ground-based observations of dust abundance over the past four decades are evaluated (Fig. 1). The simulated dust emission patterns geographically align with ground-observed dust abundance for both regions and seasons (Fig. 1a–d). Statistically significant positive correlations are widely obtained across both regions, especially over areas close to the dust sources (Fig. 1e–h). These results indicate that this model successfully captures the spatial and temporal patterns of observed dustiness.

## 2.7 Sensitivity experiments

To quantify the contribution of multiple environmental factors on East Asian and North American springtime dust emission changes, we conduct a set of sensitivity experiments that simulate controlled dust emissions. The controlled dust emissions are obtained by individually replacing the concurrent snow cover, temperature, soil moisture, hourly surface wind speed, and LAI during the controlled period with that during the baseline periods. These sensitivity experiments are conducted during the long (1980–2021) and short (2000–2021) terms as follows: in the long-term range, the time subsection is from the late 20th century (1980–2000, baseline period) to early 21st century (2001–2021, controlled period); in the short-term range, the time subsection is from 2000–2010 (baseline period) to 2011–2021 (controlled period). In this study, we use the percentage of dust emission changes between the controlled and baseline simulations to quantify the contribution of each meteorological and biological factor to the decadal changes in dust emission.

In addition, to analyze the specific contribution of ECs, we perform an additional cyclone-controlled experiment in which cyclone-affected wind speeds (Sect. 2.5) are replaced with climatological surface wind speed. This approach allows direct quantification of the contribution of ECs to near-surface wind variability and, consequently, its effect on springtime dust emission.

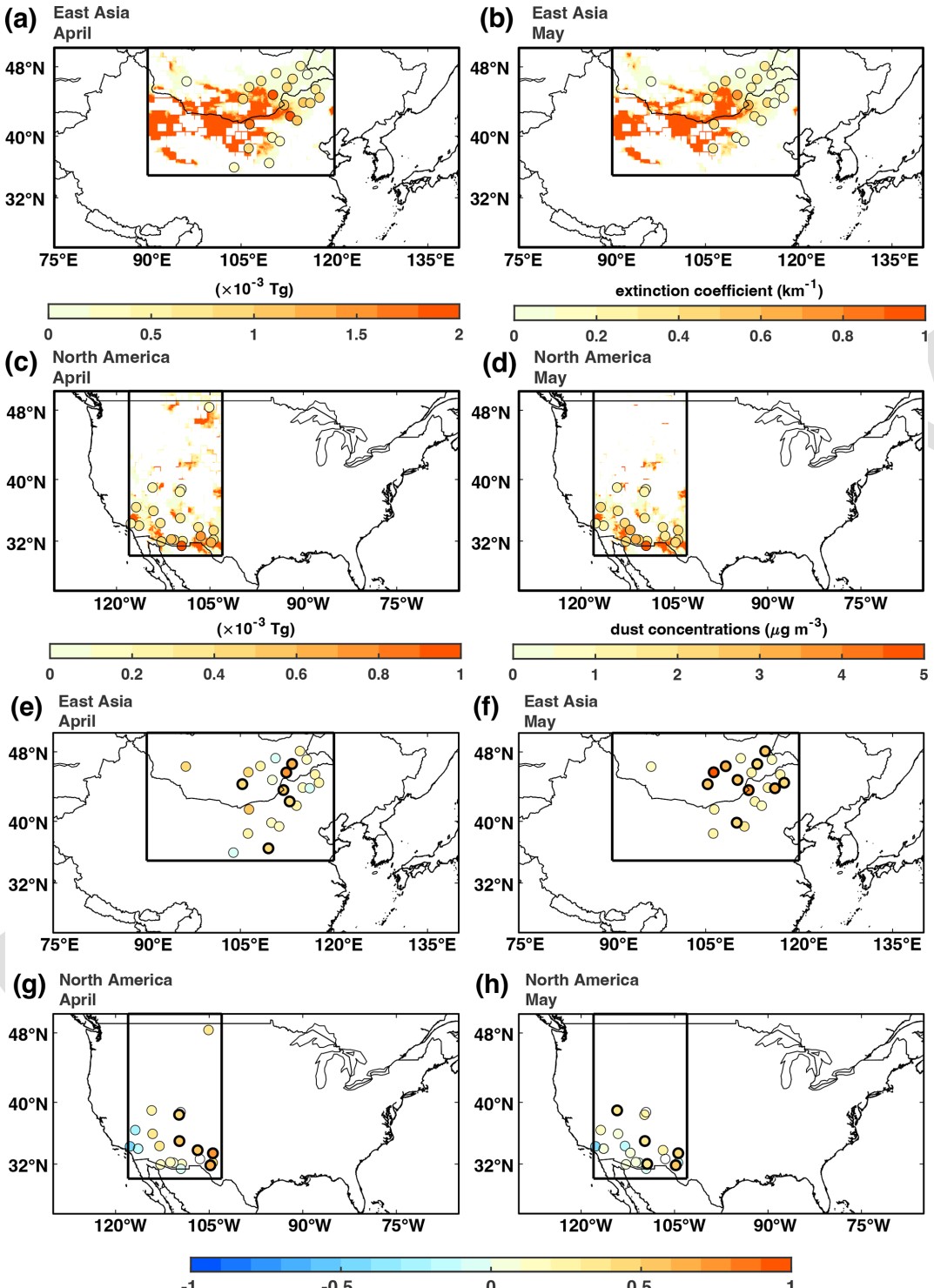

**Figure 1.** Validation of simulated dust emissions across East Asia and North America using station-based observations. Climatology of **(a, c)** April and **(b, d)** May dust emissions across **(a, b** East Asia and **(c, d)** North America during 1980–2021, with dots indicating **(a, b)** the dust aerosol extinction coefficients from duISD during 1980–2019 and **(c, d)** surface fine dust concentrations from IMPROVE during 1988–2021. Correlations between monthly **(a, b)** dust aerosol extinction coefficients from duISD during 1980–2019, **(c, d)** surface fine dust concentrations from IMPROVE during 1988–2021 and the simulated dust emission from the 0.1° grid cell covering the station in **(a, c)** April and **(b, d)** May. Dots enclosed by a black bold border indicate statistical significance at $p$ value $< 0.1$.

## 3   Results

### 3.1   Decadal variations in East Asian and North American dust emissions

Regional dust emission across East Asia mainly occurs in South Mongolia and North China, and increases by 0.240 Tg per month per decade in April during the period 1980–2021 (Fig. 2a and c). The simulated time series for East Asian dust emissions from 1980 to 2019 shows a statistically significant ($p$ values $< 0.001$, based on Spearman correlation test) positive correlation with ground-based observations (Fig. 2a). The East Asian dust emission in April shifts from a rising to declining trend after the onset of the 21st century, with a significant ($p$ values $< 0.05$, based on the Mann–Kendall trend test) reduction of 9.37 Tg per month per decade or 16.5 % per two decades from 2000 to 2021 (Fig. 2e). Consistent with this simulated decrease in April dust emission in 2000–2021, the observation dataset shows a significant ($p$ values $< 0.001$) positive correlation with the simulations, with a correlation coefficient ($r$) of 0.79 (Fig. 2a). Meanwhile, regional dust emission in North America shows a reversed multidecadal trend, with a significant ($p$ values $< 0.05$) increase of 0.406 Tg per month per decade or 23.4 % per four decades in April during the period 1980–2021 (Fig. 3c). This increase is corroborated by a significant positive correlation ($r = 0.79$, $p$ values $< 0.001$) with surface fine dust concentrations from 1988 to 2021 (Fig. 3a). However, this increase is followed by a decrease in the regional total dust emissions by 0.235 Tg per month per decade or 2.52 % per two decades in April for the period 2000–2021, which is also significantly positively correlated ($r = 0.76$, $p$ values $< 0.001$) with station-observed data (Fig. 3a and e).

In contrast, both East Asian and North American dust emissions show a consistent upward trend in May during the past four decades. The East Asian dust emission in May is estimated to have increased significantly ($p$ values $< 0.05$) by 0.937 Tg per month per decade or 5.67 % during 1980–2021; this increase accelerates to 6.22 Tg per month per decade or 11.2 % for the period 2000–2021 (Fig. 2b, d and f). This trend is supported by significant ($p$ values $< 0.001$) positive correlations with station-based observations, with correlation coefficients of 0.66 and 0.84 during past four to two decades (Fig. 2b). In North America, the regional total dust emission has increased significantly ($p$ values $< 0.1$) by 0.275 and 0.184 Tg per month per decade or 16.3 % and 12.0 % in May during 1980–2021 and 2000–2021 (Fig. 3b, d and f), respectively, consistent with the ground-based dustiness observations, which exhibit a significant positive correlation ($p$ values $< 0.001$) (Fig. 3b).

### 3.2   Influencing factors of dust emission changes since the 1980s

The contributions of several environmental variables to the decadal variations in regional dust emissions are disentangled by the sensitivity experiments. The multidecadal change in East Asian and North American springtime dust emissions during 1980–2021 have been mainly driven by variations in surface wind speed (Fig. 4a). For example, during 1980–2021, the changes in surface wind speed have made a positive contribution to dust emission increase by 10.3 % and 23.7 % across East Asia and North America, respectively, in April (Fig. 4a and e), and a corresponding regional contribution of 6.09 % and 14.5 % in May (Fig. 4c and g). During 2000 to 2021, the surface wind speed has caused a reduction in dust emissions by 22.9 % and 1.48 % across East Asia and North America in April (Fig. 4b and f) and an increment by 13.4 % and 12.2 % in May (Fig. 4d and h). As the dominant influencing factor of East Asian and North American dust emission in mid-to-late spring, near-surface wind speed shows spatio-temporally in-phase variations with dust emission. Spatially, daily maximum wind speed (Fig. 5) exhibits similar patterns of change with those in dust emissions across both regions during both the shorter and longer periods (Figs. 2 and 3).

Soil moisture constitutes the secondary control on dust emission changes in both regions and months, complementing the control of wind speed changes in a nonlinear way (Fig. 4). Despite substantial declines in soil moisture that promote dust emission potentials across both regions in all the study periods (Fig. 6), these changes are often insufficient to initiate dust emission with the absence of strong surface wind, resulting in dust emission changes that follow wind speed variations in both regions (Fig. 4). For example, April dust emissions in North America show a decreasing trend (Figs. 3e and 4f) despite continuous soil drying (Fig. 6f) during 2000–2021, primarily due to the lack of strong winds (Fig. 5f) that offsets the apparent dominance of soil moisture (Fig. 4f). By contrast, changes in vegetation exert only a minor influence on dust emission changes in two regions during the same periods (Fig. 4), likely due to lack of significant, positive trends in LAI across both regions in both April and May, especially in the longer term (Fig. 7).

### 3.3   Changes in extratropical cyclones and associated wind responsible for dust emission changes

As demonstrated in Sect. 3.2, springtime dust emissions in East Asia and North America are closely linked to wind speed and exhibit pronounced decadal variations; indeed, much of these decadal variations in the occurrence and duration of strong winds is attributable to regime shifts in ECs (Fig. 8). In East Asia, cyclone-affected strong winds predominantly shift into the longer-lasting (duration ranging from 150 to 450 h) and higher-frequency (occurring be-

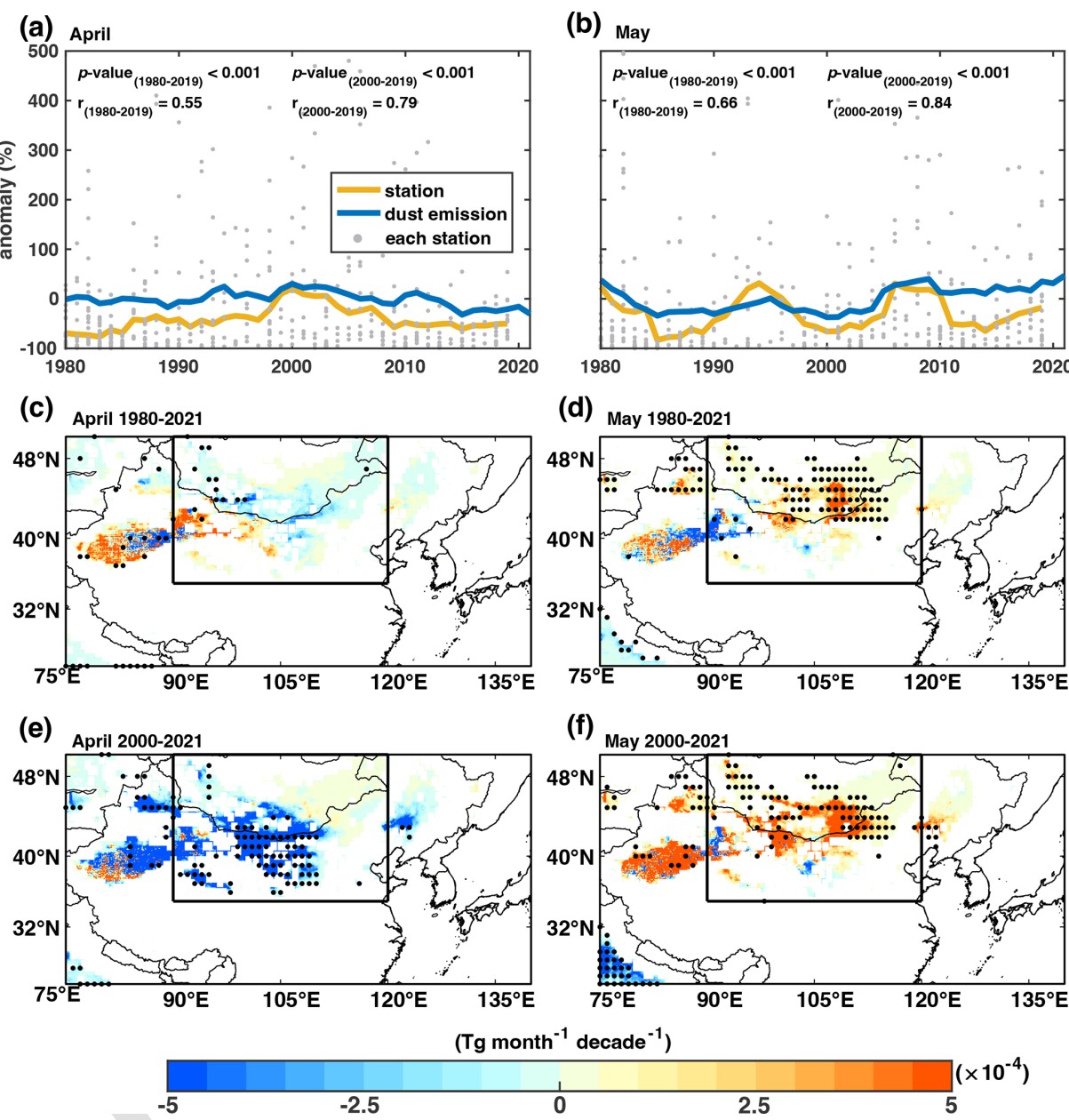

**Figure 2.** Changes in East Asian dustiness in April and May during 1980–2021 and 2000–2021. Anomaly time series of the ground-observed extinction coefficient contributed by dust aerosol, from each station (grey dot) of duISD and their median (yellow line), during 1980–2019 and dust emission anomaly (blue lines) from off-line simulation model during 1980–2021 across East Asia in **(a)** April and **(b)** May. Color shading represents the trend of simulated dust emissions (Tg per month per decade) in **(c, e)** April and **(d, f)** May for the period **(c, d)** 1980–2021 and **(e, f)** 2000–2021. Stippled areas exhibit statistically significant dust emission trends ($p$ values < 0.1, based on the Mann–Kendall trend test). Boxes denote studied dust source regions across East Asia.

tween 15 and 35 times) bins in April and May, compared with non-cyclone-affected strong winds (Fig. 8a and b). In contrast, North American cyclone-affected strong winds exhibit a less pronounced increase in both duration and frequency (Fig. 8c and d). Comparing different decades, the occurrence of longer-lasting and higher-frequency cyclone-affected strong wind ($> 6\,\mathrm{m\,s}^{-1}$) events has increased sig-

nificantly during the past four decades across East Asia and North America in both April and May (Fig. 8e and f). During the recent two decades, such cyclone-affected shift towards longer-lasting and higher-occurrence of strong winds has continued in May across East Asia (Fig. 8j), but has faded in both months across North America (Fig. 8k and l) and in April across East Asia (Fig. 8i). Meanwhile, the non-

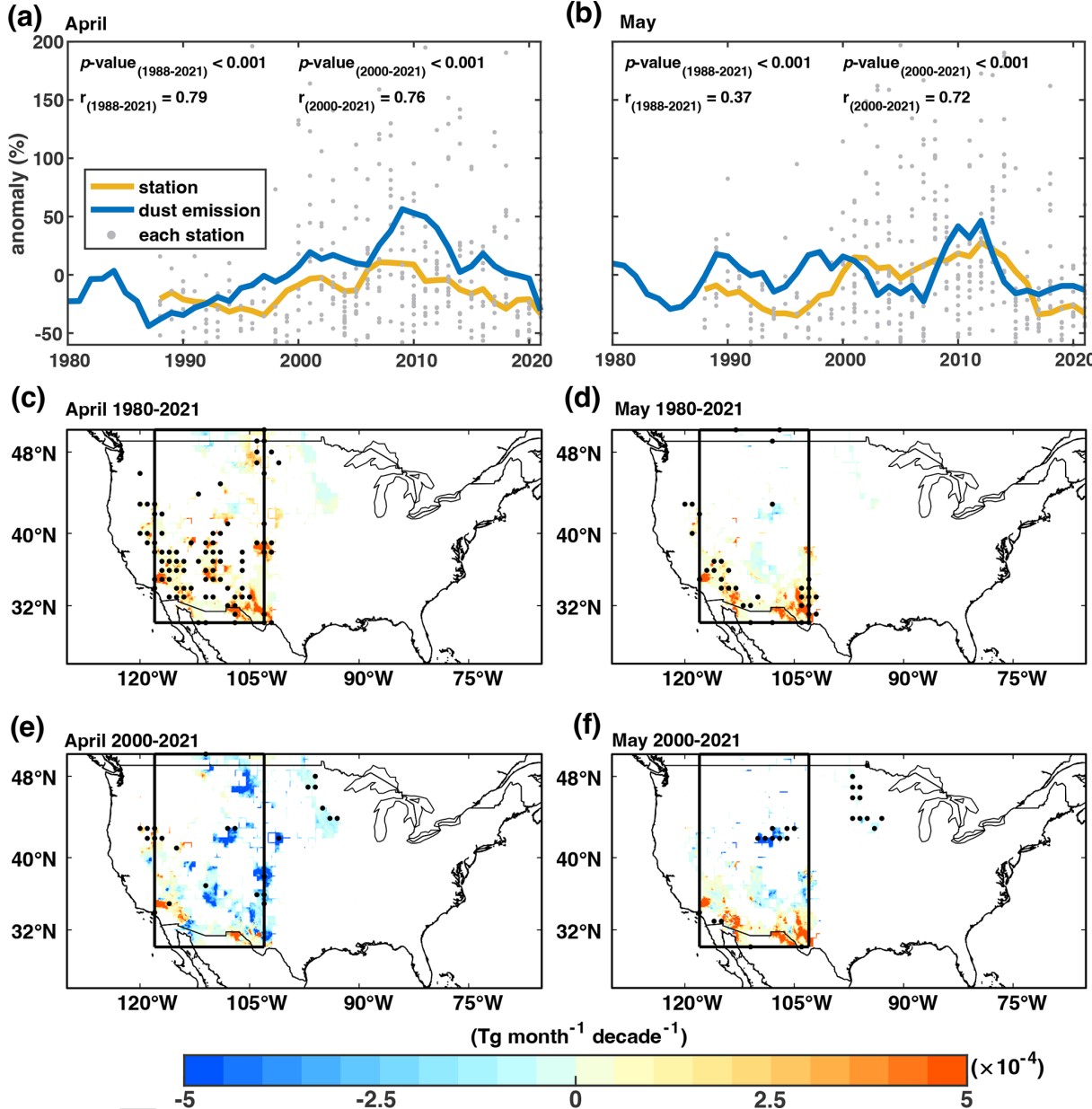

**Figure 3.** Changes in North American dustiness in April and May during 1980–2021 and 2000–2021. Anomaly time series of ground-observed surface fine dust concentrations from each station (grey dot) of IMPROVE and their median (yellow line) during 1988–2021 and dust emission anomaly (blue lines) from off-line simulation model during 1980–2021 across North America in **(a)** April and **(b)** May. Color shading represents the trend of simulated dust emissions (Tg per month per decade) in **(c, e)** April and **(d, f)** May for the period **(c, d)** 1980–2021 and **(e, f)** 2000–2021. Stippled areas exhibit statistically significant dust emission trends ($p$ values $< 0.1$, based on the Mann–Kendall trend test). Boxes denote studied dust source regions across North America. TS2

cyclone-affected strong wind events exhibit a much weaker change during the same periods (Fig. 8m–t), indicating that the decadal variations in the statistics of strong winds are primarily driven by ECs.

Furthermore, the spatiotemporal variations in wind speed are closely connected to characteristics of ECs in East Asia and North America in both April and May. According to the compilation of all cyclone events across both regions and in both months, the maximum surface wind speed within the cyclone radius shows a significant positive correlation with the central pressure and radius of ECs from 1980 to 2021 ($p$ values $< 0.001$). Next, we explore the decadal variations in wind attributable to EC characteristics.

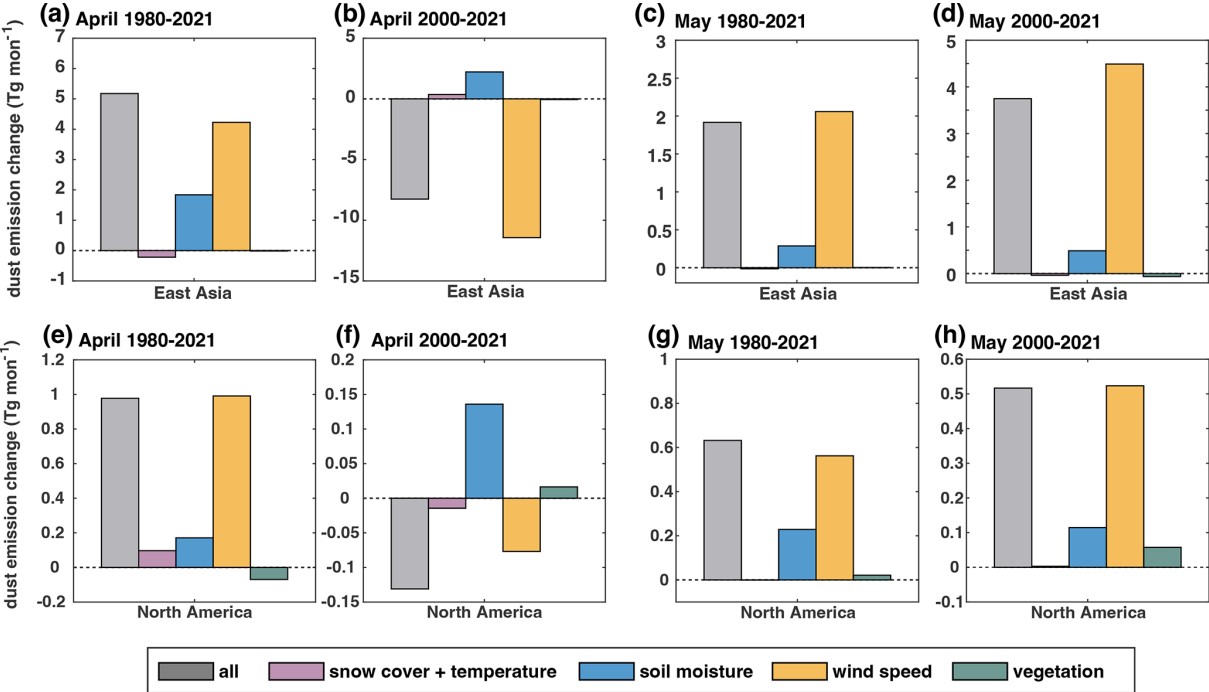

**Figure 4.** Changes in East Asian and North American dust emission (Tg per month) and the contribution of several environmental variables during 1980–2021. **(a–d)** East Asian and **(e–h)** North American dust emission change (gray, Tg per month) and the contribution of each factor (purple: snow cover fraction and land surface temperature; blue: top-layer soil moisture; yellow: near-surface wind speed; green: vegetation) in **(a, b, e, f)** April and **(c, d, g, h)** May in **(a, c, e, g)** past four decades and **(b, d, f, h)** past two decades.

In East Asia, the cumulative frequency and duration of strong wind events align closely with relative shifts among different cyclone characteristics. In April, increases in both the number of ECs and their $V_{max}$ are associated with prolonged durations of strong winds during 1980–2021 (Figs. 8e and 9a). However, in the period 2000–2021, the reduction in cyclone size counterbalanced the increase in cyclone frequency, leading to a decrease in longer-lasting strong wind events (Figs. 8i and 9a). By contrast, the expansion of cyclone radius and the increase in cyclone frequency counteracted the impact of the weakening $V_{max}$, ultimately leading to an increase in wind speed despite the reduction in $V_{max}$ during both 1980–2021 and 2000–2021 in May (Figs. 8f, j and 9b).

In North America, variations in surface wind are also explainable by changes in extratropical cyclone characteristics. In April, changes in strong wind conditions occur in conjunction with different combinations of cyclone properties, including increases in cyclone radius, frequency, and $V_{max}$ during 1980–2021, responsible for the increasing duration of cyclone-affected strong winds (Fig. 9c). During 2000–2021, the duration of cyclone-affect strong wind changes subtly due to minor changes in $V_{max}$, cyclone number, and cyclone radius (Fig. 9c). These contrasting cyclone configurations are consistent with the corresponding variability in strong winds (Fig. 8g and k). In May for the period 1980–2021, changes in

cyclone characteristics and strong winds are broadly similar to those in April over the same period (Figs. 8g, h and 9c, d). By contrast, during 2000–2021 in May, reductions in cyclone frequency and radius occur alongside an increase in $V_{max}$; the net effect is a decrease in the duration of strong winds (Figs. 8l and 9d).

Such wind speed changes associated with the regime shift in ECs have been largely responsible for the decadal variations in dust emissions from these two mid-latitude sources, with generally stronger influences across East Asia than North America (Figs. 8 and 10). According to our cyclone-controlled experiments, ECs account for 60.3 % and 38.7 % of April dust emissions in East Asia and North America, respectively, and 70.6 % and 31.5 % of May dust emissions in these two regions during 1980–2021. Similarly, during 2000–2021, ECs contribute to 60.1 % and 42.6 % of April dust emissions in East Asia and North America, respectively, and 61.9 % and 32.5 % of May dust emissions in these regions (Fig. 10). The generally lower contribution of ECs to North American dust emission is consistent with the weaker modulation of ECs on the frequency and duration of strong wind (Fig. 8a–d).

Based on the cyclone-controlled sensitivity experiments (Sect. 2.7), we further quantify the influence of extratropical cyclones on the decadal variability of dust emissions in April and May. After constraining the cyclone-affected wind

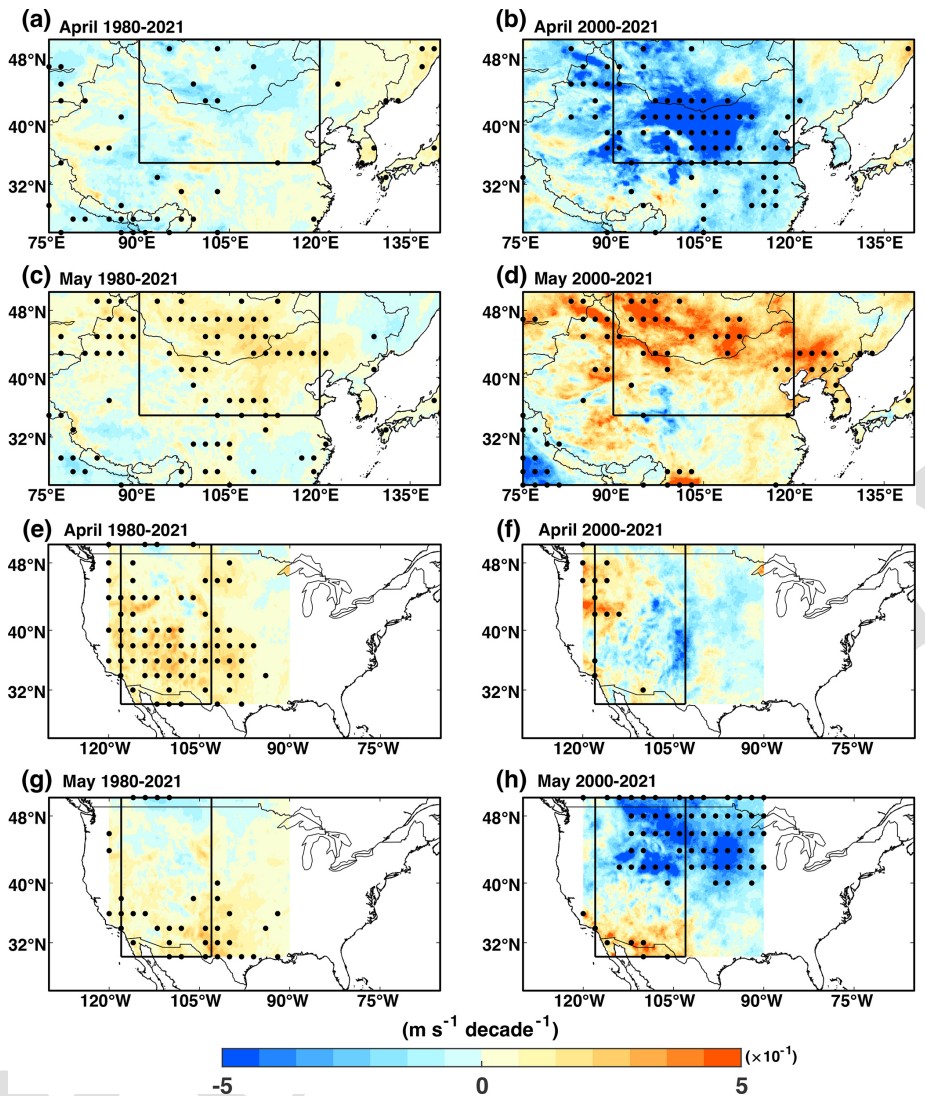

**Figure 5.** Changes in daily maximum wind speed in April and May for the period 1980–2021 and 2000–2021. Trends in **(a, b)** April and **(c, d)** May daily maximum wind speed (m s$^{-1}$ per decade) across East Asia for the period **(a, c)** 1980–2021 and **(b, d)** 2000–2021. Trends in **(e, f)** April and **(g, h)** May daily maximum wind speed (m s$^{-1}$ per decade) across North America for the period **(e, g)** 1980–2021 and **(f, h)** 2000–2021, respectively. Black dots indicate significant ($p$ values < 0.1, based on the Mann–Kendall trend test) trend.

speed to its climatological state, the decadal variability of dust emissions shows substantial changes, accompanied by a shift in the dominant environmental drivers (Fig. 11). Specifically, the magnitude of dust emission changes across both East Asia and North America is markedly reduced over the past two to four decades. The increase in East Asian dust emissions over 1980–2021 declines from 5.18 to 1.08 Tg in April, representing a reduction of 79.2 % (Figs. 4a and 11a). Similarly, in North America, the April dust emission increment over same period is reduced from 0.978 to 0.179 Tg, corresponding to a reduction of 81.7 % (Figs. 4e and 11e). In May of these four decades, nudging the cyclone-affected strong winds to their climatology leads to a reduction of 31.3 % and 37.8 % in the decadal changes of East Asian

and North American dust emission. During 2000–2021, such contribution of ECs to dust emission shrinks to 62.7 % and 58.4 % for East Asia in April and May and becomes negligible for North America in both months.

Apart from that, the dominant environmental drivers of dust emission also shift when cyclone-affected wind speeds are removed. For instance, soil moisture emerges as the primary positive contributor, accounting for 6.17 % of the East Asian dust emission increase in April during 1980–2021, while the total dust emission increased by only 6.44 % in the cyclone-controlled experiments (Fig. 11a). By contrast, the contribution of wind speed to dust emissions is reduced to merely 0.62 % after cyclone-affected winds are constrained (Fig. 11a). Naturally, such shift in the dominant environmen-

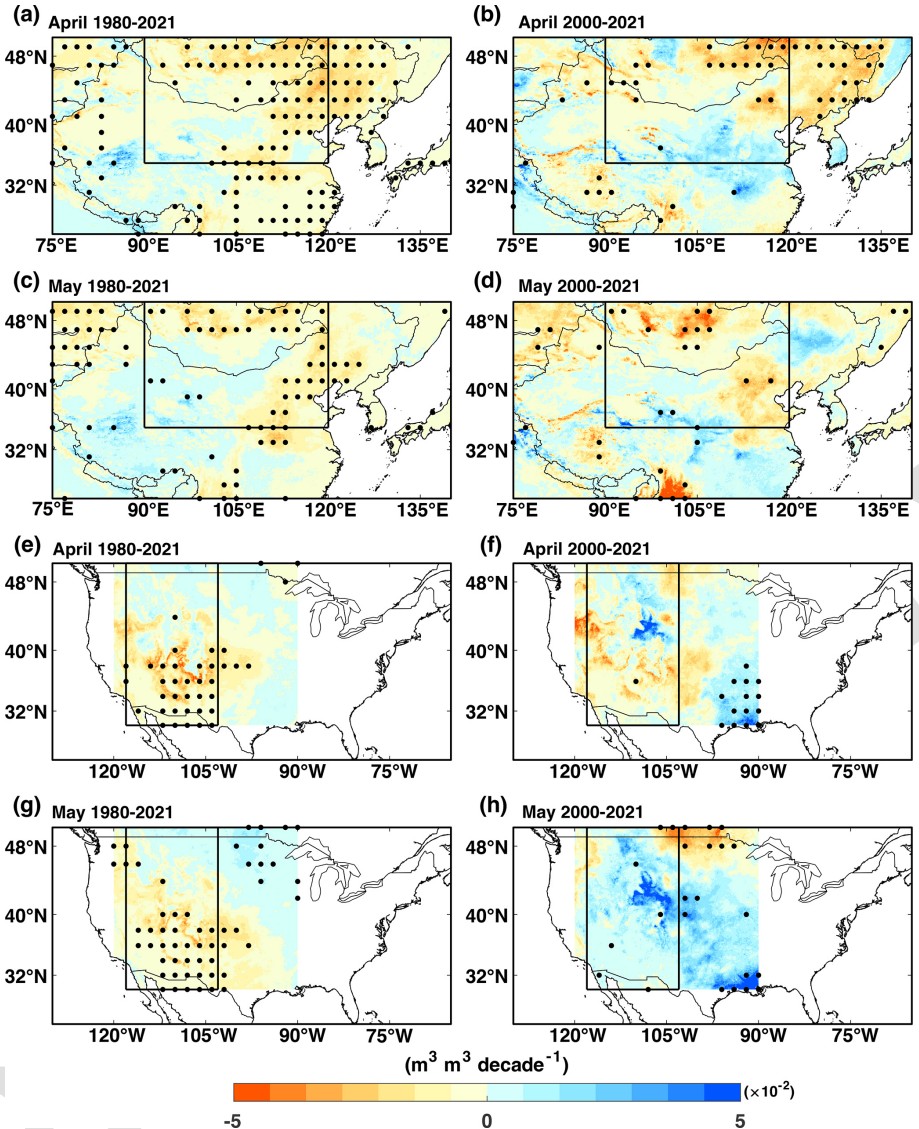

**Figure 6.** Changes in top-layer soil moisture in April and May for the period 1980–2021 and 2000–2021. Trends in **(a, b)** April and **(c, d)** May soil moisture ($m^3\,m^{-3}$ per decade) across East Asia for the period **(a, c)** 1980–2021 and **(b, d)** 2000–2021. Trends in **(e, f)** April and **(g, h)** May daily soil moisture ($m^3\,m^{-3}$ per decade) across North America for the period **(e, g)** 1980–2021 and **(f, h)** 2000–2021, respectively. Black dots indicate significant ($p$ values < 0.1, based on the Mann–Kendall trend test) trend.

tal drivers of dust emission is muted during 2000–2021, especially in North America, when and where ECs contribute negligibly to the decadal variations in dust emission.

## 4   Discussion and conclusion

Based on a suite of multi-source observational datasets and a dust emission model, we characterize the decadal variability of mid-to-late springtime dust emissions across East Asia and North America, which are primarily regulated by changes in surface wind speed and extratropical cyclone activity during the recent decades. During the past four decades, the East Asian and North American drylands exhibit a 12.7 %

and 23.4 % increase in April dust emissions and a 5.7 % and 16.3 % increase in May. During the past two decades, these two regions show a 16.5 % and 2.52 % decrease in April dust emissions and a 11.2 % and 12.0 % increase in May. Our results highlight the dominant role of surface wind speed in shaping decadal variations of dust emissions, while the frequency and intensity of extratropical cyclones exert substantial influence on wind speed variability. Collectively, these two factors constitute the primary drivers of regional total dust emission changes across East Asia and North America in the late 20th century and early 21st century. Overall, our study provides a clearer understanding of the decadal-scale variability of mid-to-late springtime dust emissions across

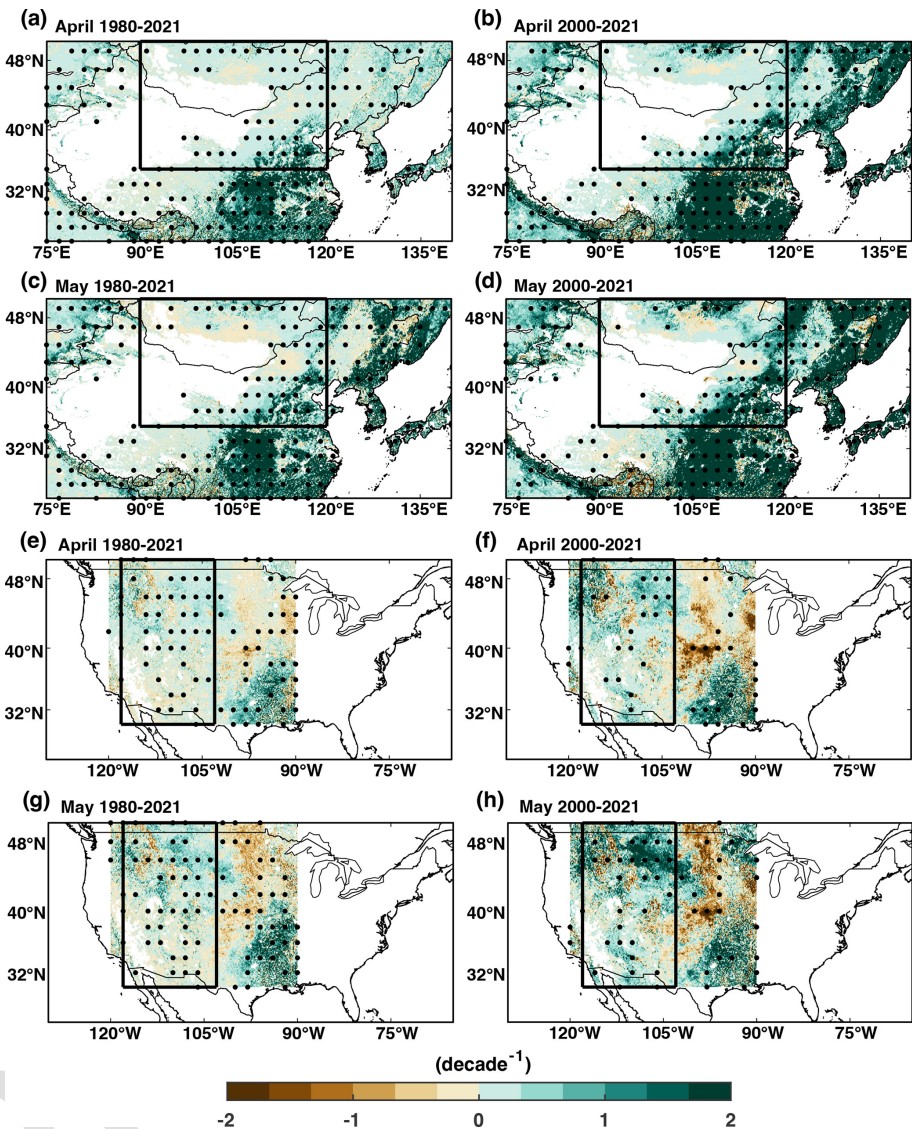

**Figure 7.** Changes in LAI in April and May for the period 1980–2021 and 2000–2021. Trends in **(a, b)** April and **(c, d)** May LAI (per decade) across East Asia for the period **(a, c)** 1980–2021 and **(b, d)** 2000–2021. Trends in **(e, f)** April and **(g, h)** May daily LAI (per decade) across North America for the period **(e, g)** 1980–2021 and **(f, h)** 2000–2021, respectively. Black dots indicate significant ($p$ values < 0.1, based on the Mann–Kendall trend test) trend. TS3

East Asia and North America, and underscores the primary roles of both surface wind speed and extratropical cyclones in modulating dust emission changes.

In this study, we demonstrate the leading influence of surface wind speed on decadal changes of dust emission. Changes in wind regimes, particularly variations in the frequency, duration, and intensity of strong wind events, play a central role in shaping long-term dust emission variability and reflect the combined influence of climate variability and climate change. Extratropical cyclones exert a strong influence on near-surface strong winds, which in turn drive dust emissions. Through their regulation of the occurrence, frequency, and duration of strong wind events, cy-

clones provide an effective dynamical linkage between large-scale atmospheric circulation and surface dust emission processes. Quantitative assessment using cyclone-controlled experiments reveals a 60 %–70 % contribution to the springtime dust emissions in East Asia and 30 %–40 % in North America, as well as a ∼ 80 % contribution to both regions' decadal variations in April dust emission and ∼ 30 % of that in May during the past four decades; whereas during the past two decades, variations in cyclone characteristics explain about ∼ 60 % of the decadal variations in April–May dust emission from East Asia but negligible to that from North America. These results support a strong dynamical coupling between cyclone-modulated near-surface winds and dust emis-

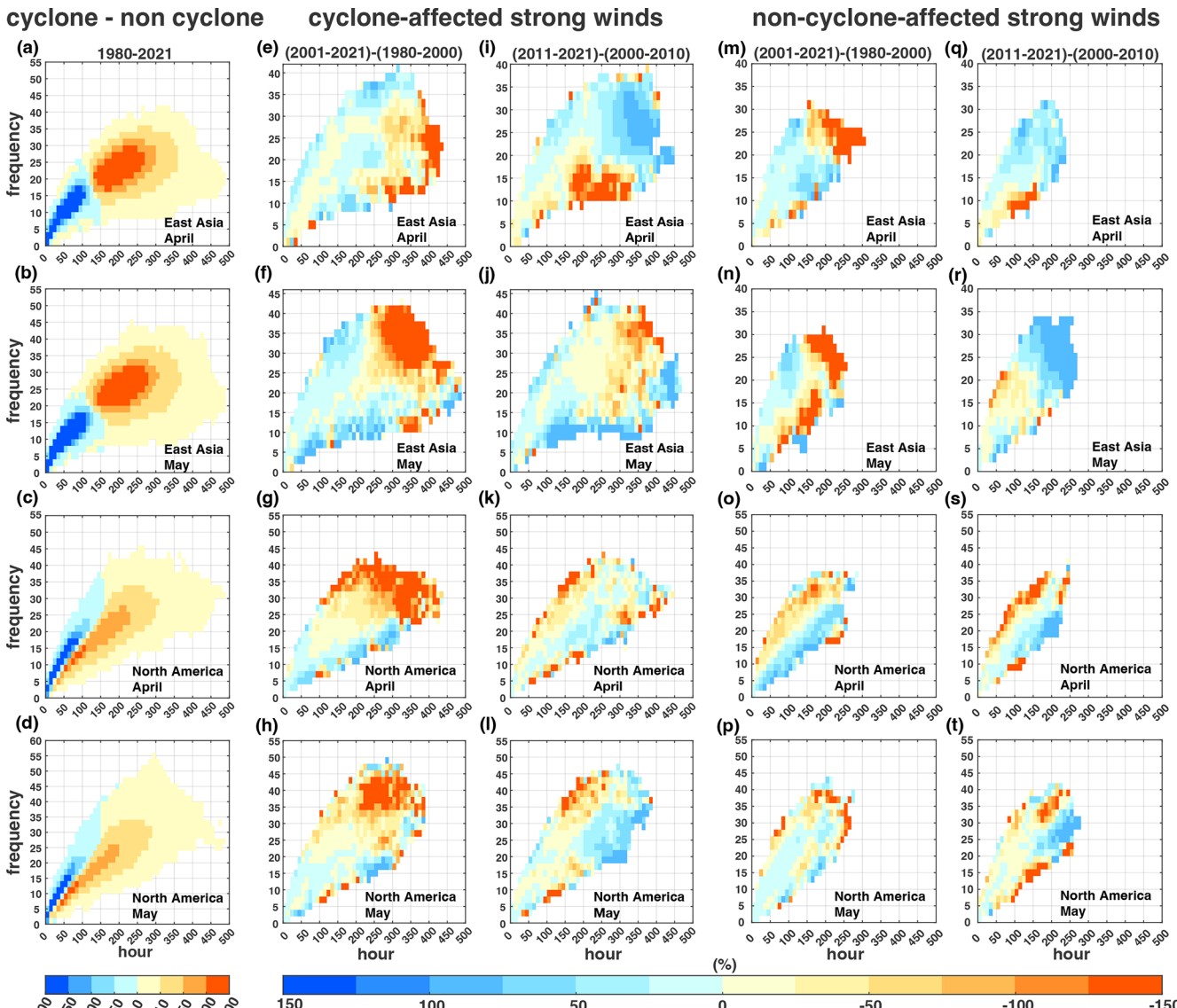

**Figure 8.** Changes in cyclone-affected strong wind across East Asia and North America in April and May. Joint probability distribution of monthly differences in total frequency and duration of cyclone-affected strong winds, subtracted by non-cyclone-affected strong winds, across the dust-emitting pixels in **(a, b)** East Asia and **(c, d)** North America in **(a, c)** April and **(b, d)** May during 1980–2021. Change rate (%) of the joint PDF of the frequency (events per month) and duration (hours per month) of cyclone-affected strong winds ($> 6\,\mathrm{m\,s^{-1}}$) to baseline periods from ERA5 hourly 10 m wind speed data across the dust-emitting pixels in **(e, f, i, j)** East Asia in **(e, i)** April and **(f, j)** May for the period **(e, f)** 1980–2021, **(i, j)** 2000–2021 and in **(g, h, k, l)** North America in **(g, k)** April and **(h, l)** May for the period **(g, h)** 1980–2021, **(k, l)** 2000–2021. **(m–t)** Same as **(e)**–**(l)** but for non-cyclone-affected strong winds.

sions across both regions in mid-to-late spring, particularly in East Asia, where the impact of extratropical cyclones is especially pronounced on the longer-lasting (duration ranging from 150 to 450 h) and higher-frequency (occurring in a range of 15 to 35 times) strong winds (Fig. 8a and b).

Beyond ECs, changes in dust emission can also be associated with changes in other synoptic-scale circulation systems, such as the Siberian High (Kang et al., 2022; Zhao et al., 2018), and meso- to small-scale processes, including

convective storms ("haboobs") (Foroutan and Pleim, 2017; Bukowski and van den Heever, 2020), nocturnal low-level jets and mountain-valley circulations (Fiedler et al., 2013; Ge et al., 2016). These processes can locally or episodically enhance near-surface winds and thereby contribute to dust emission change independently of extratropical cyclone activity.

The identified decadal changes in near-surface wind speed, along with the changing duration of strong wind events, can

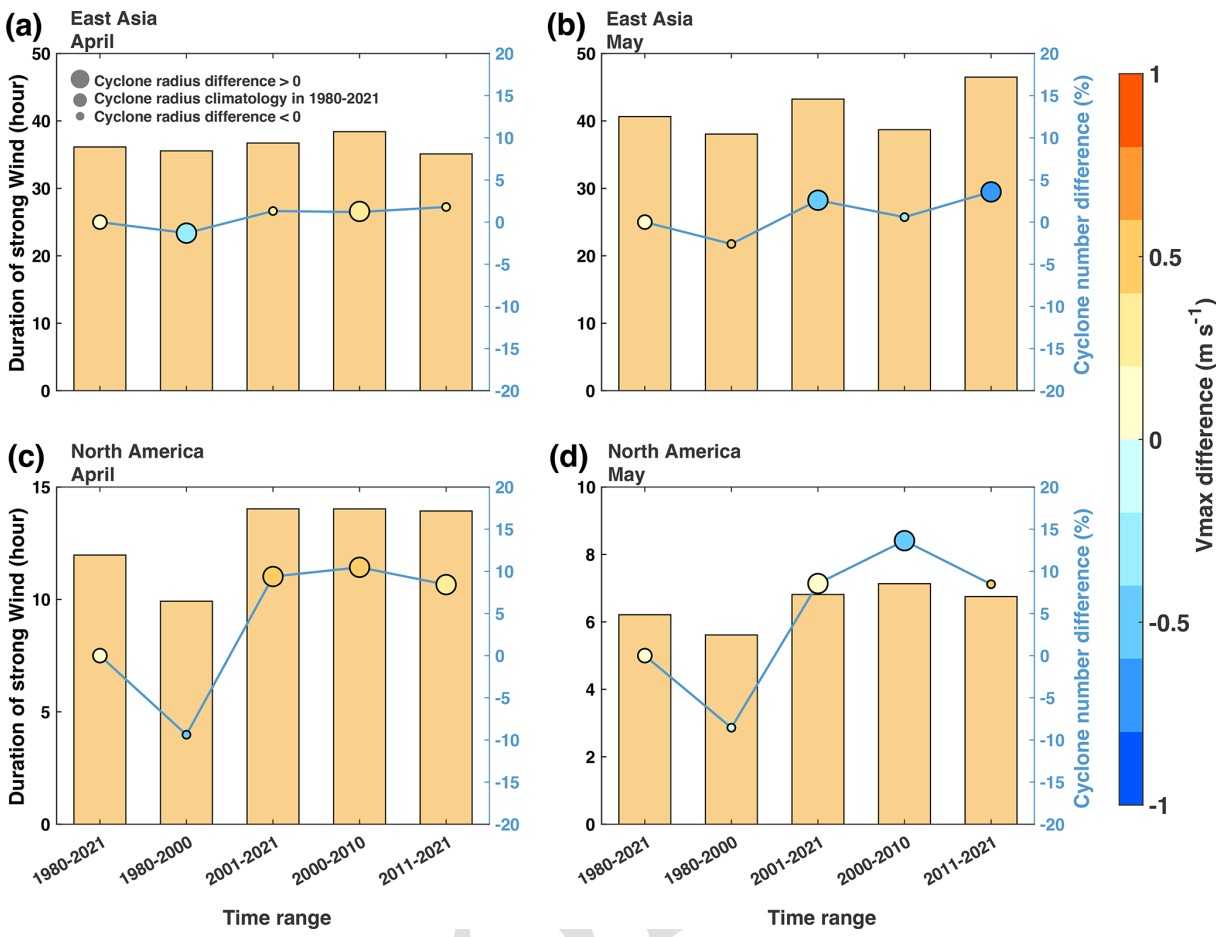

**Figure 9.** Regime shifts in extratropical cyclones across East Asia and North America during April and May across different time periods from 1980 to 2021. The monthly average duration of strong winds (hours; yellow bars) caused by extratropical cyclones in **(a, b)** East Asia and **(c, d)** North America in **(a, c)** April and **(b, d)** May, for the time periods: 1980–2021, 1980–2000, 2001–2021, 2000–2010, and 2011–2021, with reference to the left $y$ axis. The blue solid line with markers represents the difference in the monthly average number of cyclones during these periods compared to the monthly average cyclone count during the whole period 1980–2021, corresponding to the right blue $y$ axis. Marker color shows the deviation of monthly mean $V_{max}$ (m s$^{-1}$) from the 1980–2021 climatology, and marker size reflects the cyclone radius difference relative to the 1980–2021 mean.

be largely attributed to regime shifts in extratropical cyclone characteristics, including changes in cyclone frequency, intensity, and spatial extent (Figs. 8–11). In addition, these changes can be interpreted within the context of large-scale climate dynamics, including (1) the response in mid-latitude storm track processes to global warming (Shaw et al., 2016), (2) regional climate oscillations associated with large scale modes of climate oscillation, such as El Niño–Southern Oscillation (ENSO), North Atlantic Oscillation (NAO), Pacific Decadal Oscillation (PDO), Arctic Oscillation (AO) etc. (Yin et al., 2022), and (3) global surface wind stilling up to 2010 and subsequent recovery attributed to internal climate variability (Zeng et al., 2019; Wohland et al., 2021).

Compared with wind speed, land surface changes seem secondary in shaping the decadal variations in dust emission. In addition to reflecting the integrated influence of climate variability, land surface factors directly respond to climate change. For example, studies on vegetation phenology have reported an earlier greening trend across Northern Hemispheric mid-latitudes in response to early-spring warming and $CO_2$ fertilization (Fan et al., 2014; Piao et al., 2019). However, the suppressive effect associated with vegetation greening appears insufficient to offset the dominant influence of surface wind speed on dust emissions at the decadal scale (Fig. 4). Furthermore, future changes in vegetation cover depend strongly on the competing trajectories of surface temperature and soil moisture, and their role in dust emission remains uncertain (Ding et al., 2020). At the same time, non-photosynthetic vegetation present in spring over arid and semi-arid regions, such as senescent plants and crop residues, can exert a persistent suppressive effect on dust emission by modifying surface roughness and soil exposure, thereby pro-

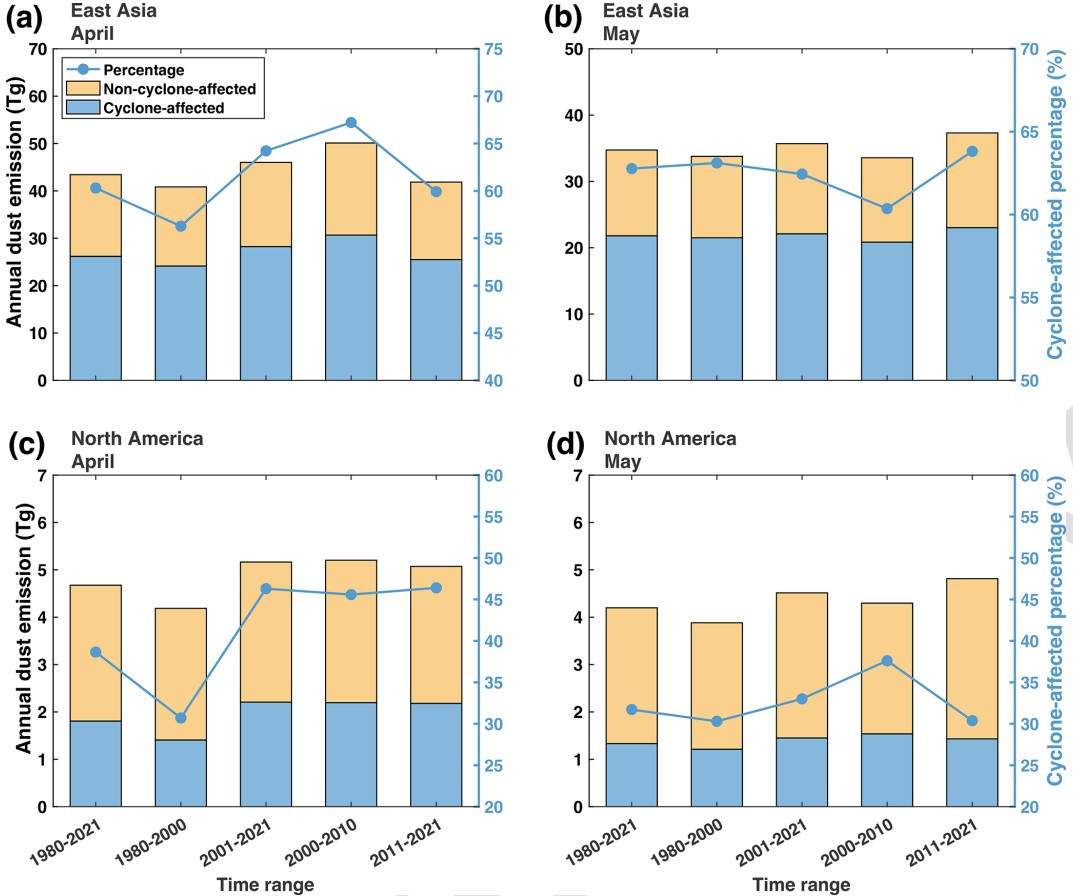

**Figure 10.** Regime shifts in extratropical cyclone-affected dust emissions (Tg) across East Asia and North America during April and May for multiple subperiods within 1980–2021. The annual average dust emissions (Tg) along the passage of extratropical cyclones (blue bars) and those unaffected by them (yellow bars) are shown for **(a, b)** East Asia and **(c, d)** North America in **(a, c)** April and **(b, d)** May, for the time periods: 1980–2021, 1980–2000, 2001–2021, 2000–2010, and 2011–2021, with reference to the left $y$ axis. The blue bars show the cyclone-affected dust emissions, defined as the difference between total dust emissions and the emissions in the cyclone-controlled experiments (i.e., total – cyclone-controlled), while the yellow bars show the emissions estimated from the cyclone-controlled experiments. The blue solid line represents the percentage of dust emissions affected by extratropical cyclones (%) over these periods, corresponding to the right $y$ axis.

viding a form of absolute but relatively stable constraint on dust emission (Huang and Foroutan, 2022).

The uncertainty in our study mainly comes from the limitations of observational datasets and dust emission model. First, due to the high temporal and spatial inhomogeneity of station observation datasets, aggregating them into a single time series leads to considerable uncertainty. Although individual station observations exhibit strong correlations with simulated dust emissions within the surrounding 0.1° grid cells (Fig. 1), the correlation between observations and simulations weakens after constructing the time series and taking the median of station anomalies. Nevertheless, the correlation remains statistically significant (Figs. 2 and 3). Second, higher albedo of cloud and land surface, in the presence of thick clouds and snow, respectively, brings challenge to satellite aerosol retrieval algorithms in the mid-latitude dust sources, preventing a more accurate quantification of

dust concentration or emission solely based on satellite data (Meng et al., 2025). Third, although the simulation from off-line dust emission model generally matches observed spatio-temporal variations, this parameterization inevitably under-represents actual physical processes, similar to all dust emission models currently being used, especially the interaction between environmental variables. For example, we estimate the area of unvegetated, wind-erosive regions within each grid by $\exp(-1 \times \text{LAI})$ (Pu and Ginoux, 2017). This parameterization, however, omits the influence of vegetation height and canopy structure on near-surface wind profile and eventually the frictional wind speed that is directly responsible for dust emission. This uncertainty in dust emission modeling will be quantified and reduced upon an expanded collection of observable data, e.g. meter-resolution vegetation structure, spatio-temporally resolved near-surface wind speed profiles,

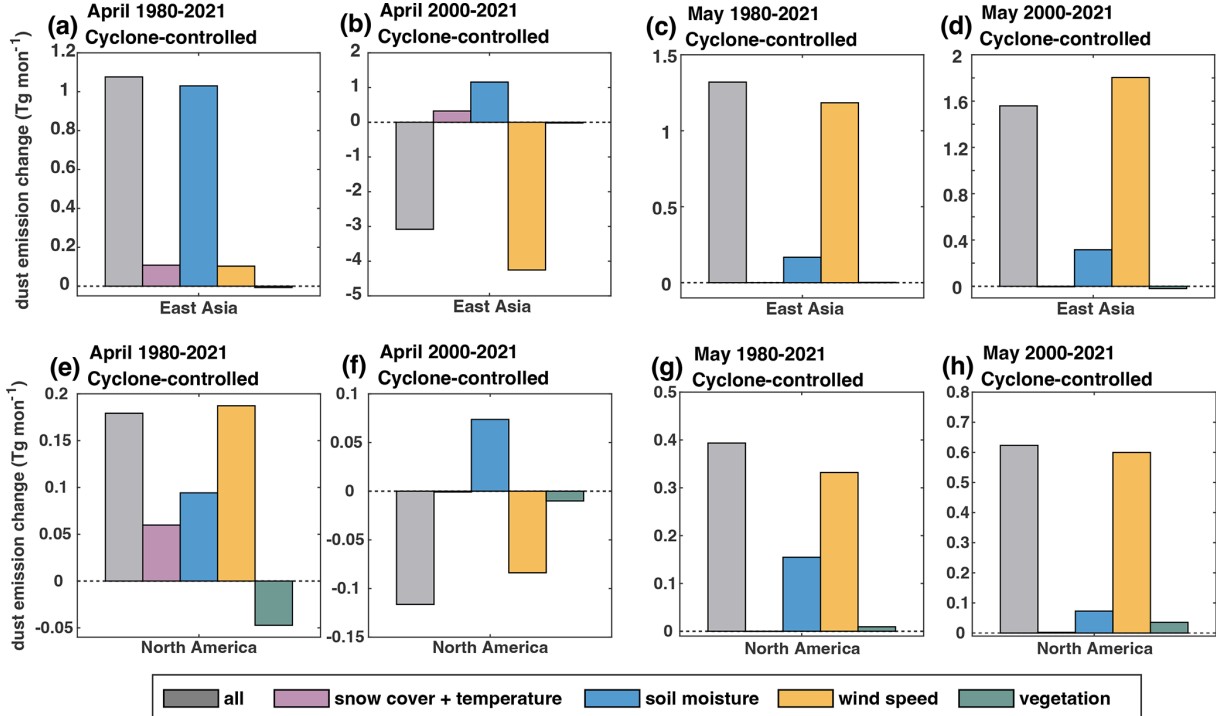

**Figure 11.** Changes in East Asian and North American dust emission (Tg per month) and the contribution of several environmental variables during 1980–2021 in cyclone-controlled experiments. Figure elements are identical to those in Fig. 4.

in conjunction with satellite measurement of dust aerosol abundance with finer spatio-temporal resolutions.

**Code availability.** The code to carry out the current analyses is available from the corresponding authors upon request.

**Data availability.** Data used in this study are all publicly available, including: the MODIS Deep Blue aerosol products acquired from the Level-1 and Atmosphere Archive and Distribution System (LAADS) Distributed Active Archive Center (DAAC), available at: https://ladsweb.modaps.eosdis.nasa.gov/ (NASA, 2024); the ERA-5 hourly climate data provided by European Centre for Medium-Range Weather Forecasts (ECMWF), available at: https://doi.org/10.24381/cds.e2161bac TS4 (Muñoz-Sabater et al., 2021; Muñoz Sabater, 2019), last accessed on 5 September 2024; the GIMMS leaf area index at a half-month temporal resolution acquired from Cao et al. (2023); the global dust Integrated Surface Database (duISD) acquired from Xi (2021); the Interagency Monitoring of Protected Visual Environments (IMPROVE) network is available at: http://vista.cira.colostate.edu/improve TS5 (IMPROVE, 2024).

**Author contributions.** YY conceived the study. YW and YY performed the analysis and wrote the initial draft. All authors contributed to the data analysis and manuscript editing.

**Competing interests.** The contact author has declared that none of the authors has any competing interests.

**Disclaimer.** Publisher's note: Copernicus Publications remains neutral with regard to jurisdictional claims made in the text, published maps, institutional affiliations, or any other geographical representation in this paper. The authors bear the ultimate responsibility for providing appropriate place names. Views expressed in the text are those of the authors and do not necessarily reflect the views of the publisher.

**Acknowledgements.** We thank Paul Ginoux for useful discussions. Computation is supported by High-performance Computing Platform of Peking University.

**Financial support.** This research has been supported by the National Natural Science Foundation of China (grant no. 42275016) and the Natural Science Foundation of Beijing Municipality (grant no. JQ23037). TS6

**Review statement.** This paper was edited by Sergio Rodríguez and reviewed by two anonymous referees.

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

## Remarks from the typesetter

TS1    Please check; a comma does not make sense here.

TS2    Please give an explanation of why this figure needs to be changed. We have to ask the handling editor for approval. Thanks.

TS3    Please give an explanation of why this figure needs to be changed. We have to ask the handling editor for approval. Thanks.

TS4    Please confirm change to DOI and the citations.

TS5    Please confirm adjusted URL (to be identical with the one in the reference list entry.

TS6    Please confirm both Acknowledgements and Financial support sections.

TS7    Please provide date of last access.

TS8    Please confirm reference list entry.