# Peer review of "Attributing the decadal variations in springtime East Asian and North American dust emission to regime shifts in extratropical cyclone 3 Yiting Wang1, Yan Yu1,2,3, Ji Nie1,2,3, Bing Pu4 4 5 6 1Department of Atmospheric and Oceanic Sciences,"

_EGUsphere, 2025_

## Referee Comment (RC2)

Main overall comment: The paper by Wang et al. provides an interesting analysis using the dust observations and modeling to explain what driving variables of dust emissions have changed in the past decades that caused the springtime dust trend. The study topic is appropriate for the ACP journal and the investigate of the dust trends' drivers is timely. The validity of the study's conclusion is based on how well the dust emission model captures the observed dust variability. An issue is that the model evaluation seems to be a little loose, based on only Fig. 1a-b and 2a-b. Instead of showing grid-level model-observation comparison, these panels show regionally aggregated time series, and there are no statistics to quantify the model performance. It is not too convincing that the dust emission model captures the observed dust variability. The ground-based dust observations and the satellite retrievals are also not largely consistent with each other, and this requires an explanation too. The authors should address the issue of model accuracy, and I suggest a major revision. I also ask a few questions regarding the interpretation of the cyclone-induced winds in the specific comments.

**Other specific comments:**

Lines 112: Some readers may not be familiar with the duISD product. The logic of the formula requires a brief explanation. What is the definition of dust frequency, and where/how were the frequency and visibility data obtained? Please describe it in the main text in 2-3 sentences.

Line 114: From Fig. 1, it seems that the duISD measurements do not agree well with the MODIS dust AOD product in Sect. 2.2. The discrepancy requires an explanation. Which one should we trust?

Line 153: Please provide the top layer soil thickness from ERA5-land.

Figs. 1 and 2: The dust emission model seems to fail to reproduce the interannual variability of the MODIS and ground-based dust, although it captures the general secular trend displayed by the observations. E.g., in Fig. 2a, the blue line has a 2010 peak, which is absent in both the satellite and the ground-based dust observations. Please discuss the reasons for such mismatch in the main text.

Lines 261-262: Why does the box not include the area of the Taklamakan desert, which is the most important desert in East Asia?

Fig. 3: Why do we see strong control by soil moisture and snow in East Asia in April but not May?

- Fig. 4: Why is soil moisture less important in North America overall?
- Fig. 5: A similar figure for ERA5-LAND soil moisture is needed.
- Line 341: A definition for "longer-lasting strong winds" is needed here.
- Fig. 7: I might have overlooked, but a dynamical explanation for why the winds are decreasing in April and increasing in May in East Asia is needed. Why does North America not experience a similar seasonality?

Line 402: It is unclear to reader what are the "non-cyclone-affected" strong winds. Please define and explain. For instance, the semi-permanent Siberian High from the north or some sea breezes from the coastal east?

Fig. 9: it is unclear to reader why this article focuses on cyclones if both cyclone and non-cyclone winds have changed and have caused the dust emissions to change across decades. Aren't there other important synoptic weather patterns that dominate surface winds over the deserts, like the Siberian high-induced cold advections and fronts?

Sect. 4: It's not very clear how your modeled dust emissions are sensitive to the prescribed extratropical cyclones. Do you have any idea if you removed all the extratropical cyclonedriven strong winds from your model, how much would your simulated dust emissions drop? A sensitivity run may help quantify the sensitivity of the simulated dust emissions to the cyclones.

Lines 449-450: How did the extratropical storm track change in the past decades and how did they change the wind direction and intensity over Mongolia and the USA, respectively?

---

## Author Comment (AC1)

Dear reviewer,

We appreciate your insightful comments and constructive suggestions. We have incorporated these suggestions in the revised manuscript. Key modifications include:

(1) Adding the cyclone-controlled sensitivity experiments to quantify the contribution of extratropical cyclones on wind speed and dust emission.
(2) Further validating the simulated dust emission from a spatiotemporal perspective based on station observational datasets.
(3) Expanding the discussion to include dust-initiating wind sources other than cyclones and the impact of non-photosynthetic vegetation on dust generation.

The line numbers refer to the clean version of the revised manuscript. We hope these modifications have strengthened our manuscript.

Yiting and Yan on behalf of all authors

Reviewer #2 (Comments for the Author):

The paper by Wang et al. provides an interesting analysis using the dust observations and modeling to explain what driving variables of dust emissions have changed in the past decades that caused the springtime dust trend. The study topic is appropriate for the ACP journal and the investigation of the dust trends' drivers is timely. The validity of the study's conclusion is based on how well the dust emission model captures the observed dust variability. An issue is that the model evaluation seems to be a little loose, based on only Fig. 1a-b and 2a-b. Instead of showing grid-level model-observation comparison, these panels show regionally aggregated time series, and there are no statistics to quantify the model performance. It is not too convincing that the dust emission model captures the observed dust variability. The ground-based dust observations and the satellite retrievals are also not largely consistent with each other, and this requires an explanation too. The authors should address the issue of model accuracy, and I suggest a major revision. I also ask a few questions regarding the interpretation of the cyclone-induced winds in the specific comments.

Reply: Thank you for your valuable suggestions regarding the validation of the simulation datasets. In the revised manuscript, we have evaluated the temporal and spatial consistency of the simulated dust emissions using station observational datasets (Fig. 1), and provided correlation coefficients and significance tests to quantify the model's performance (Fig. 2, 3).

[revised manuscript text omitted]

Other specific comments:

Lines 112: Some readers may not be familiar with the duISD product. The logic of the formula requires a brief explanation. What is the definition of dust frequency, and where/how were the frequency and visibility data obtained? Please describe it in the main text in 2-3 sentences.

Reply: Thank you for your valuable suggestions regarding the explanation of duISD in Section 2.1. We have revised this section to include a concise introduction to each factor involved in the calculation of duISD:

*"The observed extinction coefficient contributed by dust aerosol (β, km$^{-1}$) across East Asia (35 °N-50 °N, 90 °E-120 °E) is provided by global dust Integrated Surface Database (duISD) covering the period 1980-2019 (Xi, 2021). This dataset compiles about 30,000 stations globally, as collected by the National Oceanic and Atmospheric Administration (NOAA), and derives dust extinction coefficient from visibility observations as follows:*

$$\beta = \frac{3.9}{V} \times f \, , \tag{1}$$

*where β is a measure of the extinction coefficient caused by dust particles, V is the harmonic mean visibility associated with dust events, and f is the dust frequency (%) given by:*

$$f = \frac{N_{du}}{N_{ww}} \times 100\% , \qquad\qquad (2)$$

*Here, $N_{du}$ is the number of reported dust events, and $N_{ww}$ is the total number of weather reports (ww) during a given time period (Shao et al., 2013; Kurosaki and Mikami, 2003). Weather reports from manned stations are categorized by the World Meteorological Organization (WMO) under Code Table 4677, with priority codes ranging from 00 (lowest) to 99 (highest), indicating the visual perception of weather phenomena during the observation period. Dust events are ranked within the fog (40-49) and precipitation (50-99) weather groups and are identified by the following numeric codes: ww = 06-09, 30-35, 98.”* (lines 112-127)

Line 114: From Fig. 1, it seems that the duISD measurements do not agree well with the MODIS dust AOD product in Sect. 2.2. The discrepancy requires an explanation. Which one should we trust?

Reply: Thank you for raising this issue. We find that duISD measurements agree with dust emission model much better than MODIS dust AOD on the monthly scale. After examining the sampling of MODIS DOD, we realized that over half of days in a typical April or May are not sampled by MODIS, likely due to the cloud contamination problem. Therefore, in the revised manuscript, we remove the MODIS DOD-related analysis in validation. We have addressed the uncertainty in satellite remote sensing in discussion:

*“Second, higher albedo of cloud and land surface, in the presence of thick clouds and snow, respectively, brings challenge to satellite aerosol retrieval algorithms in the mid-latitude dust sources, preventing a more accurate quantification of dust concentration or emission solely based on satellite data (Meng et al., 2025).”* (lines 582-585)

Yet, this sampling issue of MODIS DOD is mitigated after compiling over two decades of data. Therefore, it is still used as the basis for the observational, climatological dust source function in our off-line dust emission model.

Line 153: Please provide the top layer soil thickness from ERA5-land.

Reply: Thank you, we have clarified the top lay soil thickness from ERA5-LAND:

*“To investigate the change in dust emissions and the contribution of several environmental variables in April and May, we analyze the 6-hourly snow cover fraction (%), top layer soil moisture (0-7 cm, $m^3$ $m^{-3}$), land surface temperature (K) and hourly 10-m wind speed (m $s^{-1}$) from the European Centre for Medium-Range Weather Forecasts Reanalysis v5-Land (ERA5-LAND, referred to ERA5 hereafter) during 1980-2021.”* (lines 171-175)

Figs. 1 and 2: The dust emission model seems to fail to reproduce the interannual variability of the MODIS and ground-based dust, although it captures the general secular trend displayed by the observations. E.g., in Fig. 2a, the blue line has a 2010 peak, which is absent in both the satellite and the ground-based dust observations. Please discuss the reasons for such mismatch in the main text.

Reply: Thank you for your valuable suggestions on the validation. Due to the high temporal and spatial variability of station datasets, aggregating them into a single time series introduces uncertainty, resulting in differing time series characteristics between station observational datasets and dust emission simulation. In the revised manuscript, we have updated the processing of station data by selecting only those stations within a 0.1° radius where simulated dust emissions are nonzero for further analysis. This approach yielded a statistically significant positive correlation between station observations and simulated dust emissions

(Fig. 2, 3). We performed a more formal spatiotemporal validation of the dust emission model as in the response to your major comment.

Lines 261-262: Why does the box not include the area of the Taklamakan desert, which is the most important desert in East Asia?

Reply: Thank you for questioning our study region. We choose to focus on Gobi Desert mainly because of two reasons. The first reason is the apparently limited influence of ECs on Taklamakan desert, as reported by Mu and Fiedler (2025) among other studies. As stated by Mu and Fiedler (2025), different meteorological processes contribute to dust emission and affect the subsequent atmospheric transport in the deserts. Dust emissions in the Gobi Desert has been reported as mainly driven by ECs, namely Mongolian Cyclone; whereas dust emissions in the Taklamakan Desert are driven by multiple physical mechanisms, e.g., the breakdown of Nocturnal Low-Level Jets and dry convection in heat lows. Furthermore, as also mentioned by Mu and Fiedler (2025) among other studies, due to topographical differences, Gobi dust has larger potential for long-range transport compared with Taklamakan dust and more direct influence on the downwind, populated region. Therefore, we focus our current analysis on ECs' influence in the Gobi Desert. In the revised manuscript, we have added a brief explanation of our study region in the Introduction. We appreciate your suggestion and will expand our analysis to Taklamakan desert in the near future.

"*Apart from the influence on the natural environment, extreme dust activities from the Gobi Desert and Southwest United States also impair atmospheric visibility, air quality, and human health across downwind regions, including populated areas in China and the United States (Gui et al., 2022; Hashizume et al., 2020; Yang et al., 2015).*" (lines 36-40)

"*Moreover, intense dust storm events that frequently occur in April-May over the Gobi Desert and Southwest United States are often modulated by extratropical cyclones, and associated storm tracks or frontal systems (Lukens et al., 2018; Guo et al., 2017).*" (lines 83-86)

Fig. 3: Why do we see strong control by soil moisture and snow in East Asia in April but not May?

Reply: Thank you for raising the issue of soil moisture and snow. We apologize for the minor error in East Asian dust emission simulation, which does not affect the final conclusion regarding the relationship between wind speed, extratropical cyclones, and dust emissions. In the revised manuscript, soil moisture continues to have a stronger contribution to East Asian dust emissions in April than in May, particularly during the 1980-2021 period. This is primarily due to the more pronounced drying trend (Fig. 6):

"*Soil moisture constitutes the secondary control on dust emission changes in both regions and months, complementing the control of wind speed changes in a nonlinear way (Fig. 4). Despite substantial declines in soil moisture that promote dust emission potentials across both regions in all the study periods (Fig. 6), these changes are often insufficient to initiate dust emission with the absence of strong surface wind, resulting in dust emission changes that follow wind speed variations in both regions (Fig. 4). For example, April dust emissions in North America show a decreasing trend (Figs. 3e, 4f) despite continuous soil drying (Fig. 6f) during 2000-2021, primarily due to the lack of strong winds (Fig. 5f) that offsets the apparent dominance of soil moisture (Fig. 4f).*" (lines 342-350)

Fig. 4: Why is soil moisture less important in North America overall?

Reply: Thank you for raising the issue of soil moisture. After correcting the computational error, it is not obvious that soil moisture is less important in North America. In the revised manuscript, we have explained the contribution of soil moisture to the changes in dust emissions in both East Asia and North America, and

analyzed the changes in soil moisture across these two regions for the periods 1980-2021 and 2000-2021 (Fig. 6).

*"Soil moisture constitutes the secondary control on dust emission changes in both regions and months, complementing the control of wind speed changes in a nonlinear way (Fig. 4). Despite substantial declines in soil moisture that promote dust emission potentials across both regions in all the study periods (Fig. 6), these changes are often insufficient to initiate dust emission with the absence of strong surface wind, resulting in dust emission changes that follow wind speed variations in both regions (Fig. 4). For example, April dust emissions in North America show a decreasing trend (Figs. 3e, 4f) despite continuous soil drying (Fig. 6f) during 2000-2021, primarily due to the lack of strong winds (Fig. 5f) that offsets the apparent dominance of soil moisture (Fig. 4f)."* (lines 342-350)

Fig. 5: A similar figure for ERA5-LAND soil moisture is needed.

Reply: Thank you for your constructive suggestion, we have analyzed the changes in soil moisture across East Asia and North America for the periods 1980-2021 and 2000-2021 (Fig. 6).

Line 341: A definition for "longer-lasting strong winds" is needed here.

Reply: Thank you for your valuable suggestion. In the revised manuscript, we have provided definitions for "longer-lasting strong winds" and "higher-frequency strong winds" in section 3.3:

*"Comparing different decades, the occurrence of longer-lasting and higher-frequency cyclone-affected strong wind ($> 6 \ m \ s^{-1}$) events has increased significantly during the past four decades across East Asia and North America in both April and May (Fig. 8e-f)."* (lines 392-395)

Fig. 7: I might have overlooked, but a dynamical explanation for why the winds are decreasing in April and increasing in May in East Asia is needed. Why does North America not experience a similar seasonality?

Reply: Thank you for your comment. The variation in wind speed can be partially attributed to the influence of cyclones. We have discussed the combined effects of extratropical cyclones on wind speed variation in section 3.3:

*"In East Asia, the cumulative frequency and duration of strong wind events align closely with relative shifts among different cyclone characteristics. In April, increases in both the number of ECs and their Vmax are associated with prolonged durations of strong winds during 1980-2021 (Figs. 8e, 9a). However, in the period 2000-2021, the reduction in cyclone size counterbalanced the increase in cyclone frequency, leading to a decrease in longer-lasting strong wind events (Figs. 8i, 9a). By contrast, the expansion of cyclone radius and the increase in cyclone frequency counteracted the impact of the weakening Vmax, ultimately leading to an increase in wind speed despite the reduction in Vmax during both 1980-2021 and 2000-2021 in May (Figs. 8f, j, 9b).*

*In North America, variations in surface wind are also explainable by changes in extratropical cyclone characteristics. In April, changes in strong wind conditions occur in conjunction with different combinations of cyclone properties, including increases in cyclone radius, frequency, and Vmax during 1980-2021, responsible for the increasing duration of cyclone-affected strong winds (Fig. 9c). During 2000-2021, the duration of cyclone-affect strong wind changes subtly due to minor changes in Vmax, cyclone number, and cyclone radius (Fig. 9c). These contrasting cyclone configurations are consistent with the corresponding variability in strong winds (Fig. 8g, k). In May for the period 1980-2021, changes in cyclone characteristics and strong winds are broadly similar to those in April over the same period (Figs.*

*8g, h, 9c, d). By contrast, during 2000-2021 in May, reductions in cyclone frequency and radius occur alongside an increase in Vmax; the net effect is a decrease in the duration of strong winds (Figs. 8l, 9d)."* (lines 409-428)

Line 402: It is unclear to reader what are the "non-cyclone-affected" strong winds. Please define and explain. For instance, the semi-permanent Siberian High from the north or some sea breezes from the coastal east?

Reply: Thank you for your suggestion. In the revised manuscript, we have provided definitions for "cyclone-affected strong winds" and "non-cyclone-affected strong winds" in section 2.5 and explained other processes that may influence wind speed in the "Discussion and conclusion" section:

*"To quantify the contribution of ECs to surface wind speed across East Asia and North America, we define all surface wind speeds and strong-wind (> 6 m s$^{-1}$) events that occur within the radial domain of each extratropical cyclone as cyclone-affected winds and cyclone-affected strong winds. Conversely, winds and strong winds outside this domain are classified as non-cyclone-affected winds and non-cyclone-affected strong winds, respectively. The spatial extent of each cyclone is determined following Schenkel et al. (2017) as the radial distance from the cyclone center at which the azimuthal-mean 10-m wind speed equals a critical wind speed threshold. Following previous studies (Pérez-Alarcón et al., 2021; Pérez-Alarcón et al., 2024), we test several thresholds (2, 4, 6, 8, 10, and 12 m s$^{-1}$) and adopt 6 m s$^{-1}$, which both aligns with our definition of strong winds and provides the most consistent results. All points within this radius are considered to be influenced by the cyclone."* (lines 200-210)

*"Beyond ECs, changes in dust emission can also be associated with changes in other synoptic-scale circulation systems, such as the Siberian High (Kang et al., 2022; Zhao et al., 2018), and meso- to small-scale processes, including convective storms ("haboobs") (Foroutan and Pleim, 2017; Bukowski and Van Den Heever, 2020), nocturnal low-level jets and mountain-valley circulations (Fiedler et al., 2013; Ge et al., 2016). These processes can locally or episodically enhance near-surface winds and thereby contribute to dust emission change independently of extratropical cyclone activity."* (lines 543-549)

Fig. 9: It is unclear to reader why this article focuses on cyclones if both cyclone and non-cyclone winds have changed and have caused the dust emissions to change across decades. Aren't there other important synoptic weather patterns that dominate surface winds over the deserts, like the Siberian high-induced cold advections and fronts?

Reply: Thank you for your constructive comment. We agree that other important synoptic and smaller-scale weather events are responsible for dust emission over these arid regions. Here we are specifically interested in ECs because they have been reported to contribute a large portion of springtime dust emission in the Gobi Desert (e.g. Mu and Fiedler (2025)) and ECs have shown substantial decadal variability (e.g. Shaw et al. (2016)). These pieces of knowledge motivate use to quantify the contribution of ECs to the decadal variations in dust emission. We hope our analysis has quantitatively connected the decadal variability in ECs to that in the mid-latitude springtime dust emission, as summarized in the discussion section.

*"Extratropical cyclones exert a strong influence on near-surface strong winds, which in turn drive dust emissions. Through their regulation of the occurrence, frequency, and duration of strong wind events, cyclones provide an effective dynamical linkage between large-scale atmospheric circulation and surface dust emission processes. Quantitative assessment using cyclone-controlled experiments reveals a 60-70% contribution to the springtime dust emissions in East Asia and 30-40% in North America, as well as a ~80% contribution to both regions' decadal variations in April dust emission and ~30% of that in May during the past four decades; whereas during the past two decades, variations in cyclone characteristics explain about ~60% of the decadal variations in April-May dust emission from East Asia but negligible to that from North America. These results support a strong dynamical coupling between cyclone-modulated near-surface*

*winds and dust emissions across both regions in mid-to-late spring, particularly in East Asia, where the impact of extratropical cyclones is especially pronounced on the longer-lasting (duration ranging from 150 to 450 hours) and higher-frequency (occurring in a range of 15 to 35 times) strong winds (Fig. 8a, b)."* (lines 527-541)

However, in addition to cyclones, many other factors also influence dust emission through their impact on wind speed. We have briefly discussed these factors in the "Discussion and Conclusion" section:

*"Beyond ECs, changes in dust emission can also be associated with changes in other synoptic-scale circulation systems, such as the Siberian High (Kang et al., 2022; Zhao et al., 2018), and meso- to small-scale processes, including convective storms ("haboobs") (Foroutan and Pleim, 2017; Bukowski and Van Den Heever, 2020), nocturnal low-level jets and mountain-valley circulations (Fiedler et al., 2013; Ge et al., 2016). These processes can locally or episodically enhance near-surface winds and thereby contribute to dust emission change independently of extratropical cyclone activity."* (lines 543-549)

Sect. 4: It's not very clear how your modeled dust emissions are sensitive to the prescribed extratropical cyclones. Do you have any idea if you removed all the extratropical cyclone-driven strong winds from your model, how much would your simulated dust emissions drop? A sensitivity run may help quantify the sensitivity of the simulated dust emissions to the cyclones.

Reply: Thank you very much for your valuable suggestions on the quantification of extratropical cyclones (ECs) contribution! We have added a cyclone-controlled experiments to clarify the effect of ECs on dust emission across East Asia and North America.

The cyclone-controlled sensitivity experiments are introduced as:
*"In addition, to analyze the specific contribution of ECs, we perform an additional cyclone-controlled experiment in which cyclone-affected wind speeds (section 2.5) are replaced with climatological surface wind speed. This approach allows direct quantification of the contribution of ECs to near-surface wind variability and, consequently, its effect on springtime dust emission."* (lines 270-273)

The quantification of ECs' contribution on dust emission is outlined in Abstract:
*"Specifically, ECs are responsible for 60-70% of the April-May total dust emissions in East Asia and 30-40% of that in North America; meanwhile, ECs explain a larger portion of the decadal variations in April dust emission from East Asia (up to ~80%), compared with May and from North America."* (lines 25-28)

And discussed in detail in Section 3.3:
*"Such wind speed changes associated with the regime shift in ECs have been largely responsible for the decadal variations in dust emissions from these two mid-latitude sources, with generally stronger influences across East Asia than North America (Figs. 8 and 10). According to our cyclone-controlled experiments, ECs account for 60.3% and 38.7% of April dust emissions in East Asia and North America, respectively, and 70.6% and 31.5% of May dust emissions in these two regions during 1980-2021. Similarly, during 2000-2021, ECs contribute to 60.1% and 42.6% of April dust emissions in East Asia and North America, respectively, and 61.9% and 32.5% of May dust emissions in these regions (Fig. 10). The generally lower contribution of ECs to North American dust emission is consistent with the weaker modulation of ECs on the frequency and duration of strong wind (Fig. 8a-d).*

*Based on the cyclone-controlled sensitivity experiments (section 2.7), we further quantify the influence of extratropical cyclones on the decadal variability of dust emissions in April and May. After constraining the cyclone-affected wind speed to its climatological state, the decadal variability of dust emissions shows substantial changes, accompanied by a shift in the dominant environmental drivers (Fig. 11). Specifically, the magnitude of dust emission changes across both East Asia and North America is markedly reduced over*

*the past two to four decades. The increase in East Asian dust emissions over 1980-2021 declines from 5.18 Tg to 1.08 Tg in April, representing a reduction of 79.2% (Figs. 4a, 11a). Similarly, in North America, the April dust emission increment over same period is reduced from 0.978 Tg to 0.179 Tg, corresponding to a reduction of 81.7% (Figs. 4e, 11e). In May of these four decades, nudging the cyclone-affected strong winds to their climatology leads to a reduction of 31.3% and 37.8% in the decadal changes of East Asian and North American dust emission. During 2000-2021, such contribution of ECs to dust emission shrinks to 62.7% and 58.4% for East Asia in April and May and becomes negligible for North America in both months.*

*Apart from that, the dominant environmental drivers of dust emission also shift when cyclone-affected wind speeds are removed. For instance, soil moisture emerges as the primary positive contributor, accounting for 6.17% of the East Asian dust emission increase in April during 1980-2021, while the total dust emission increased by only 6.44% in the cyclone-controlled experiments (Fig. 11a). By contrast, the contribution of wind speed to dust emissions is reduced to merely 0.62% after cyclone-affected winds are constrained (Fig. 11a). Naturally, such shift in the dominant environmental drivers of dust emission is muted during 2000-2021, especially in North America, when and where ECs contribute negligibly to the decadal variations in dust emission."* (lines 468-501)

Lines 449-450: How did the extratropical storm track change in the past decades and how did they change the wind direction and intensity over Mongolia and the USA, respectively?

Reply: Thank you for your question. If our understanding of the literature is correct, as a driving force for the extratropical strong wind and dust emission, storm track is another phrasing for extratropical cyclone (e.g. Guo et al. (2017)), although dynamically they have different emphases. Please correct us if our understanding is inaccurate.

We have clarified the terminology used in the manuscript in the introduction section:

*"Moreover, intense dust storm events that frequently occur in April-May over the Gobi Desert and Southwest United States are often modulated by extratropical cyclones, and associated storm tracks or frontal systems (Lukens et al., 2018; Guo et al., 2017)."* (lines 83-86)

---

## Author Comment (AC2)

Dear reviewer,

We appreciate your insightful comments and constructive suggestions. We have incorporated these suggestions in the revised manuscript. Key modifications include:

(1) Adding the cyclone-controlled sensitivity experiments to quantify the contribution of extratropical cyclones on wind speed and dust emission.
(2) Further validating the simulated dust emission from a spatiotemporal perspective based on station observational datasets.
(3) Expanding the discussion to include dust-initiating wind sources other than cyclones and the impact of non-photosynthetic vegetation on dust generation.

The line numbers refer to the clean version of the revised manuscript. We hope these modifications have strengthened our manuscript.

Yiting and Yan on behalf of all authors

Reviewers' comments:

Reviewer #1 (Comments for the Author):

This manuscript by Yiting Wang et al. present a solid and well-documented investigation into the decadal variability of springtime dust emissions across East Asia and North America, emphasizing the role of extratropical cyclone regimes. The authors combine multi-source observations and modeling to bridge the gap between regional and synoptic-scale processes. The topic is timely and of high relevance to the atmospheric and climate research community. I believe it is well-suited for publication in ACP, pending clarification and some revisions on several methodological and interpretative aspects for potential improvements.

General comments

In the data validation section, only the trend consistency between the simulation results and the observed data is compared, and no quantitative validation indicators (such as correlation coefficient, root mean square error, etc.) are provided. Supplementing these quantitative indicators can more intuitively reflect the simulation accuracy of the model.

Reply: Thank you for your valuable suggestions regarding the validation of the simulation datasets. In the revised manuscript, we have evaluated the temporal and spatial consistency of the simulated dust emissions using station observational datasets (Fig. 1), and provided correlation coefficients and significance tests to quantify the model's performance (Fig. 2, 3).

*"To assess the reliability of the off-line dust emission model over East Asia and North America during April and May, spatial distributions and temporal correlations between simulated dust emissions and ground-based observations of dust abundance over the past four decades are evaluated (Fig. 1). The simulated dust emission patterns geographically align with ground-observed dust abundance for both regions and seasons (Fig. 1a-d). Statistically significant positive correlations are widely obtained across both regions, especially over areas close to the dust sources (Fig. 1e-h). These results indicate that this model successfully captures the spatial and temporal patterns of observed dustiness."* (lines 238-245)

*"The East Asian dust emission in April shifts from a rising to declining trend after the onset of the 21st century, with a significant (p-values < 0.05, based on the Mann-Kendall trend test) reduction of 9.37 Tg*

*month$^{-1}$ decade$^{-1}$ or 16.5% per two decades from 2000 to 2021 (Fig. 2e). Consistent with this simulated decrease in April dust emission in 2000-2021, the observation dataset shows a significant (p-values < 0.001) positive correlation with the simulations, with a correlation coefficient (r) of 0.79 (Fig. 2a). Meanwhile, regional dust emission in North America shows a reversed multidecadal trend, with a significant (p-values < 0.05) increase of 0.406 Tg month$^{-1}$ decade$^{-1}$ or 23.4% per four decades in April during the period 1980-2021 (Fig. 3c). This increase is corroborated by a significant positive correlation (r = 0.79, p-values < 0.001) with surface fine dust concentrations from 1988 to 2021 (Fig. 3a). However, this increase is followed by a decrease in the regional total dust emissions by 0.235 Tg month$^{-1}$ decade$^{-1}$ or 2.52% per two decades in April for the period 2000-2021, which is also significantly positively correlated (r = 0.76, p-values < 0.001) with station-observed data (Fig. 3a, e)."* (lines 281-293)

The authors attribute the May dust emission increase to longer-lasting strong winds, but the respective contributions of cyclone-induced and non-cyclone winds are not quantitatively separated. A more explicit comparison between Figures 7 and 9 could clarify how much of the wind-driven dust increase is attributable to cyclone activity.

Reply: Thank you very much for your valuable suggestions on the quantification of extratropical cyclones (ECs) contribution! We have added a cyclone-controlled experiments to clarify the effect of ECs on dust emission across East Asia and North America.

The cyclone-controlled sensitivity experiments are introduced as:
*"In addition, to analyze the specific contribution of ECs, we perform an additional cyclone-controlled experiment in which cyclone-affected wind speeds (section 2.5) are replaced with climatological surface wind speed. This approach allows direct quantification of the contribution of ECs to near-surface wind variability and, consequently, its effect on springtime dust emission."* (lines 270-273)

The quantification of ECs' contribution on dust emission is outlined in Abstract:
*"Specifically, ECs are responsible for 60-70% of the April-May total dust emissions in East Asia and 30-40% of that in North America; meanwhile, ECs explain a larger portion of the decadal variations in April dust emission from East Asia (up to ~80%), compared with May and from North America."* (lines 25-28)

And discussed in detail in Section 3.3:
*"Such wind speed changes associated with the regime shift in ECs have been largely responsible for the decadal variations in dust emissions from these two mid-latitude sources, with generally stronger influences across East Asia than North America (Figs. 8 and 10). According to our cyclone-controlled experiments, ECs account for 60.3% and 38.7% of April dust emissions in East Asia and North America, respectively, and 70.6% and 31.5% of May dust emissions in these two regions during 1980-2021. Similarly, during 2000-2021, ECs contribute to 60.1% and 42.6% of April dust emissions in East Asia and North America, respectively, and 61.9% and 32.5% of May dust emissions in these regions (Fig. 10). The generally lower contribution of ECs to North American dust emission is consistent with the weaker modulation of ECs on the frequency and duration of strong wind (Fig. 8a-d).*

*Based on the cyclone-controlled sensitivity experiments (section 2.7), we further quantify the influence of extratropical cyclones on the decadal variability of dust emissions in April and May. After constraining the cyclone-affected wind speed to its climatological state, the decadal variability of dust emissions shows substantial changes, accompanied by a shift in the dominant environmental drivers (Fig. 11). Specifically, the magnitude of dust emission changes across both East Asia and North America is markedly reduced over the past two to four decades. The increase in East Asian dust emissions over 1980-2021 declines from 5.18 Tg to 1.08 Tg in April, representing a reduction of 79.2% (Figs. 4a, 11a). Similarly, in North America, the April dust emission increment over same period is reduced from 0.978 Tg to 0.179 Tg, corresponding to a reduction of 81.7% (Figs. 4e, 11e). In May of these four decades, nudging the cyclone-affected strong winds*

*to their climatology leads to a reduction of 31.3% and 37.8% in the decadal changes of East Asian and North American dust emission. During 2000-2021, such contribution of ECs to dust emission shrinks to 62.7% and 58.4% for East Asia in April and May and becomes negligible for North America in both months.*

*Apart from that, the dominant environmental drivers of dust emission also shift when cyclone-affected wind speeds are removed. For instance, soil moisture emerges as the primary positive contributor, accounting for 6.17% of the East Asian dust emission increase in April during 1980-2021, while the total dust emission increased by only 6.44% in the cyclone-controlled experiments (Fig. 11a). By contrast, the contribution of wind speed to dust emissions is reduced to merely 0.62% after cyclone-affected winds are constrained (Fig. 11a). Naturally, such shift in the dominant environmental drivers of dust emission is muted during 2000-2021, especially in North America, when and where ECs contribute negligibly to the decadal variations in dust emission."* (lines 468-501)

Through sensitivity experiments and LAI trend analysis, this study demonstrates that "the interdecadal changes in vegetation cover contribute minimally to dust emission," and the conclusion is reliable. However, it is important to note that this conclusion only focuses on "the contribution of vegetation changes." In contrast, the background sand-fixing effect of vegetation itself (such as the continuous inhibitory effect of stable vegetation cover on dust) falls under the category of "absolute contribution," which has not been directly quantified by the current experimental design. It is recommended to supplement some explanations in the discussion section

Reply: Thank you for raising the vegetation and surface temperature issue. In the revised manuscript, we have discussed the contribution of non-photosynthetic vegetation on dust emission:

*"At the same time, non-photosynthetic vegetation present in spring over arid and semi-arid regions, such as senescent plants and crop residues, can exert a persistent suppressive effect on dust emission by modifying surface roughness and soil exposure, thereby providing a form of absolute but relatively stable constraint on dust emission (Huang and Foroutan, 2022)."* (lines 570-574)

The study clearly identifies the significant impact of extratropical cyclones on near-surface strong winds but fails to elaborate on how cyclone regime shifts specifically regulate the frequency and duration of local strong winds. It is recommended to supplement the analysis of correlations between key cyclone parameters (e.g., central pressure gradient, vorticity distribution, and interaction between influence range and local topography) and near-surface wind fields to enhance the physical logic coherence of cyclone changes, strong wind variations, and dust emission changes.

Reply: Thank you for your constructive issue. We have analyzed the cyclone characteristics provided by the Cyclone TRACKing framework (CyTRACK, section 2.5), including central pressure and radius of extratropical cyclone. We take your advice and add the connection between EC characteristics and surface wind:

*"Furthermore, the spatiotemporal variations in wind speed are closely connected to characteristics of ECs in East Asia and North America in both April and May. According to the compilation of all cyclone events across both regions and in both months, the maximum surface wind speed within the cyclone radius shows a significant positive correlation with the central pressure and radius of ECs from 1980 to 2021 (p-values < 0.001). Next, we explore the decadal variations in wind attributable to EC characteristics."* (lines 402-407)